



**Sensitivity of different BVOC emission schemes in WRF-Chem(v3.6)**
**to vegetation distributions and its impacts over East China**
[1]Mingshuai Zhang, [1,2]Chun Zhao*, [1]Yuhan Yang, [1]Qiuyan Du, [3]Yonglin Shen, [1]Shengfu
Lin, [4]Dasa Gu
[1]School of Earth and Space Sciences, University of Science and Technology of China,
Hefei, China
[2]CAS Center for Excellence in Comparative Planetology, University of Science and T
echnology of China, Hefei, China
[3]School of Geography and Information Engineering, China University of
Geosciences, Wuhan 430074, China
[4]Division of Environment and Sustainability, Hong Kong University of Science and
Technology, Clear Water Bay, Hong Kong SAR, China

20         *Corresponding author:      Chun Zhao (chunzhao@ustc.edu.cn)

**Key points:**
1. Modeling performance of BVOC and its impact over East China using different
versions (v1.0, v2.0, v3.0) of Model of Emissions of Gases and Aerosols from Nature
(MEGAN) in WRF-Chem(v3.6) are examined and documented.
2. Three versions of MEGAN show different sensitivity to vegetation distributions and
simulate different seasonal variations of BVOC emissions over East China.
3. Temperature-dependent factor dominates the seasonal change of activity factor in all
three versions of MEGAN, while the different response to the change of leaf area index
determines the difference among the three versions in seasonal variation of BVOC
emissions.
4. The surface ozone concentration can be significantly affected by BVOC emissions
over East China, but the impact is sensitive the MEGAN versions.





## Abstract

Biogenic volatile organic compounds (BVOCs) simulated by current air quality and climate models still have large uncertainties, which can influence atmosphere chemistry and secondary pollutant formation over East China. These uncertainties are generally resulted from two sources. One is from different biogenic emission schemes coupled in model, representing for different treatments of physical and chemistry progresses during the emissions of BVOCs. The other is from the biased distribution of vegetation types over a specific region. In this study, the version of WRF-Chem updated by the University of Science and Technology of China (USTC version of WRF-Chem) from the public WRF-Chem(v3.6) is used. The modeling results over East China with different versions (v1.0, v2.0, v3.0) of Model of Emissions of Gases and Aerosols from Nature (MEGAN) in WRF-Chem are examined and documented. Sensitivity experiments with these three versions of MEGAN and two vegetation datasets are conducted to investigate the difference of three MEGAN versions in modeling biogenic VOCs and its dependence on the vegetation distributions. The experiments are also conducted for spring (April) and summer (July) to examine the seasonality of the modeling results. The results indicate that MEGANv3.0 simulates the largest amount of biogenic isoprene emissions over East China. The different performance among MEGAN versions is primarily due to their different treatments of applying emission factors and vegetation types. In particular, the results highlight the importance of considering sub-grid vegetation fraction in estimating BVOCs emissions. Among all activity factors, temperature-dependent factor dominates the seasonal change of activity factor in all three versions of MEGAN, while the different response to the leaf area index (LAI) change determines the difference among the three versions in seasonal variation of BVOC emissions. The simulated surface ozone concentration due to BVOCs can be significantly different among the experiments with three versions of MEGAN, which is mainly due to their impacts on surface VOCs and NOx concentrations. This study suggests that there is still large uncertain range in modeling BVOCs and their impacts on photochemistry and ozone production. More accurate vegetation distribution and measurements of biogenic emission flux and species concentration are needed to evaluate the model performance and reduce the uncertainties.


## 1. Introduction

Volatile organic compounds (VOCs) in the atmosphere are from biogenic and anthropogenic sources. Previous studies have indicated that biogenic emission is the dominant source of VOCs, accounting for about 90% of total emissions at global scale (Guenther et al., 1995). Biogenic VOCs (BVOCs) plays a critical role in atmosphere chemistry because some species such as isoprene and monoterpenes are reactive, and can participate in atmospheric photochemical reactions. Therefore, BVOCs could have significant impact on the formation of ozone and secondary organic aerosol (SOA) and ultimately air quality and climate change (Pierce et al., 1998; Carslaw et al., 2000; Poisson et al., 2000; Zhang et al., 2000; Carlton et al., 2009; Brown et al., 2013; Hantson et al., 2017). Among the BVOCs species, isoprene is one of the key identified species that dominates the BVOCs emissions. Global estimation also shows that biogenic isoprene emission is approximately half of total BVOCs emissions (Guenther et al., 2012).

Due to the importance of BVOCs for atmospheric environment, progress has been made extensively in modeling BVOCs emission and its impacts regionally and globally over the past several decades (Geron et al., 1994; Guenther et al., 1995; Niinemets et al., 1999; Arneth et al., 2007). BVOCs emissions are normally estimated with numerical schemes, such as the Seasonal Isoprene synthase Model-Biochemical Isoprenoid biosynthesis Model (SIM-BIM) (Lehning et al., 2001; Zimmer et al., 2003), the Biogenic Emission Inventory System (BEIS)(Pierce et al., 1998), the Global Biosphere Emissions and Interactions System (GloBEIS3) (Yarwwod et al., 2002), the semi-empirical BVOC emission model (seBVOC) (Stewart et al., 2003), and the Model of Emissions of Gases and Aerosols from Nature (MEGAN) (Guenther, 2006; Guenther et al., 2012; Zhao et al., 2016; Jiang et al., 2018). MEGAN is one of the widely used emission schemes for estimating BVOCs emissions under different environmental conditions, and has been coupled with multiple chemical transport models to include the contributions of BVOCs to the variations of air pollutants (Levis et al., 2003; Yang et al., 2011; Ghude et al., 2013; Situ et al., 2013; Tie et al., 2013; Li and Xie, 2014;



Forkel et al., 2015; Kota et al., 2015; Liu et al., 2018; Wu et al., 2020). However, there
still remain larger uncertainties in the estimation of BVOCs emission with MEGAN,
due to the uncertain emission rates of some compounds, the limited knowledge of
environmental activity factors controlling the BVOCs emissions, the accuracy of
vegetation distributions, and etc. (Guenther, 2013).

WRF-Chem (Weather Research and Forecasting model coupled with Chemistry)

is an online coupled meteorology and chemistry model that can simulate meteorology
fields and chemical species simultaneously (Grell et al., 2005; Fast et al., 2006). The
MEGAN scheme is widely used for estimating biogenic emissions online with WRF-
Chem (Jiang et al., 2012a; Wang et al., 2015; Abdi-Oskouei et al., 2018; Wei et al., 2018;
Arghavani et al., 2019; Safronov et al., 2019; Visser et al., 2019; Li et al., 2020; Yin et
al., 2020). The public versions (v4.2 and older) of WRF-Chem include the first
MEGAN version (referred to as MEGANv1.0 hereafter) (Guenther et al., 1995) and the
second version (referred to as MEGANv2.0 hereafter) (Guenther, 2006). The first
version is an earlier scheme with simple canopy treatment and chemical mechanism,
considering only the environmental effects from light and temperature on emission flux,
and therefore is mainly used in previous studies (e.g., (Guenther et al., 1996; Derognat
et al., 2003) but not often in recent studies. Comparatively, MEGANv2.0 is more
commonly used for calculating the BVOC emissions with WRF-Chem recently (Geng
et al., 2011; Jiang et al., 2012b; Zhang et al., 2015; Zhou et al., 2017) due to its treatment
of additional chemical compounds and plant types for emissions. It also considers more
complex environmental controlling processes. MEGANv2.1 (Guenther et al., 2012)
was recently coupled with WRF-Chem embedded in the CLM4 land surface scheme
(Zhao et al., 2016), so that MEGAN obtains the meteorological fields that are calculated
online and the consistent vegetation types from the land surface scheme. Although all
these three MEGAN versions were coupled in WRF-Chem and used for estimating
BVOCs emissions, so far the difference among these MEGAN versions in terms of
modeling BVOCs emission and its impacts in WRF-Chem is not examined and
documented.

With the rapid increase in economic development during the past several decades,



East China has become the most prosperous and developed region of China's economy.
More and more air pollutants and precursors are emitted into the atmosphere over the
region. Previous studies have found that BVOCs play important roles on air pollutant
production over East China (e.g., (Han et al., 2005; Wei et al., 2007; Wang et al., 2008;
Fu et al., 2010; Zheng et al., 2010; Li et al., 2015a; Li et al., 2015b). Tie et al. (2013)
found that the ozone formation was strongly VOC-limited in Shanghai of East China
and its production could partly attributed to the biogenic emission of isoprene. Jiang et
al. (2012b) investigated the impacts of local biogenic and anthropogenic emissions to
the daytime mean ozone mixing ratios over East China using WRF-Chem with
MEGANv2.0. Geng et al. (2011) applied WRF-Chem with MEGANv2.0 for studying
the effect of isoprene on ozone formation in Shanghai, and they found that the BVOCs
from the major forest surrounded have significant impact on ozone production through
two different mechanisms. Li et al. (2017a) employed WRF-Chem with MEGANv2.0
to estimate the relative contribution of biogenic and anthropogenic sources to ozone
concentration over East China, and concluded that the BVOCs contributed significantly
to the background ozone concentration. Wang et al. (2019) founded that the ozone
concentration in south of Shanghai can be enhanced significantly due to the mixing of
the emissions of BVOCs from the forest and precursors from the ships.
Since the WRF-Chem model with different MEGAN versions has been widely used
for studying the impacts of BVOCs on air quality over East China while the
performance of different MEGAN versions in WRF-Chem has not been examined, this
study aims to investigated the difference of MEGAN versions in terms of modeling
BVOCs, focusing on biogenic isoprene, and its impact on ozone concentration over
East China. This study updates the MEGANv2.1 coupled by Zhao et al. (2016) to the
latest version MEGANv3.0 (see details in Section 2.2), and analyzes the difference of
WRF-Chem modeling results with MEGANv1.0, MEGANv2.0, and MEGANv3.0.
Numerical experiments are conducted for April and July of 2015 to reflect the seasonal
variation of biogenic isoprene emissions and its potential impacts. In order to examine
the different sensitivities of MEGAN versions in WRF-Chem to vegetation
distributions, two land-use datasets are adopted in this study, which are USGS24
(United States Geological Survey 24 categories classification) and MODIS2015 (a new
dataset derived from the satellite retrievals in this study representing the land-use
condition of 2015). The paper is organized as following. Section 2 describes the
numerical experiments and methods. The results and discussions are presented in
Section 3. A summary is provided in Section 4.

**2. Methodology**
2.1.WRF-Chem
The version of WRF-Chem updated by University of Science and Technology of
China (USTC version of WRF-Chem) is used in this study. Compared with the publicly
released version, this USTC version of WRF-Chem includes some additional functions
such as the MEGAN scheme implemented in the land surface model (Zhao et al., 2013a;
Zhao et al., 2013b; Zhao et al., 2014; Zhao et al., 2016). The configuration of WRF-
Chem in this study is similar to that used by (Zhao et al., 2016). In brief, the CBM-Z
photochemical mechanism (Zaveri and Peters, 1999) is selected to simulate the gas-
phase chemistry that contains 55 prognostic species and 134 reactions. The photolysis
rates is computed by the Fast-J radiation parameterization (Wild et al., 2000), and the
Yonsei University (YSU) scheme (Hong et al., 2006) is for planetary boundary layer
(PBL) parameterization. All of the WRF-Chem simulations use the Morrison two-
moment scheme (Morrison et al., 2009) for cloud physics, the Monin–Obukhov
similarity theory (Paulson, 1970) for surface layer, the Kain–Fritsch scheme (Kain,
2004) to simulate sub-grid scale clouds and precipitation and the rapid radiative transfer
parameterization (RRTMG) for both longwave and shortwave radiation (Iacono et al.,

2008).


2.2 MEGAN implemented in WRF-Chem
MEGAN is a widely used scheme for calculating biogenic emissions from
terrestrial system to atmosphere with the impact from different environmental
conditions, such as radiation, temperature, soil moisture, and leaf area. Three versions
of MEGAN online coupled with WRF-Chem are used in this study, MEGANv1.0,



MEGANv2.0, and MEGANv3.0 that is updated from MEGANv2.1 as implemented by
(Zhao et al., 2016) according to the changes made by Jiang et al. (2018) and the
technical description of CLM4.0 (Oleson et al., 2010).
MEGAN in WRF-Chem estimates biogenic emission $(F_i)$ of different chemical
compounds $(i)$ based on emission factors $(\varepsilon_i)$ ($\mu$g m$^{-2}$h$^{-1}$), activity factors $(\gamma_i)$ that
is controlled by environmental conditions, and the lost and production rate within the
plant canopy $(\rho)$.
$F_i = \varepsilon_i \times \gamma_i \times \rho$                        (1)
Where $\varepsilon_i$ is a PFT weighted value that is calculated by PFT specific emission factor
$\varepsilon_{i,j}$ and grid box area coverage fraction $f_{PFT(j)}$ of PFT$(j)$, i.e., $\varepsilon_i = \sum \varepsilon_{i,j} f_{PFT(j)}$ and
$\gamma_i$ is the product of each activity factor such as leaf-level photosynthetic photon flux
density (PPFD)$(\gamma_P)$, temperature$(\gamma_t)$, leaf area index (LAI) $(\gamma_{LAI})$ and leaf age$(\gamma_a)$,
i.e.,$\gamma_i = \gamma_{LAI}\gamma_P\gamma_t\gamma_a$
MEGANv1.0 is the first model version coupled in WRF-Chem. It considers only
the response of emission to radiation and temperature. The mechanism of
environmental impact is very simple compared with the later versions. For emission
factors, MEGANv1.0 follows the land surface scheme with 24 land use types and
prescribes emission factor for each land-use type (Fig. 2). It groups the 24 land-use
types into the 6 plant categories (urban or bare soil, agriculture, grassland, deciduous
forest, mixed forest, and other natural land) for calculating biogenic emission activity
factor.
Guenther (2006) introduced MEGANv2.0 that is a major update from the previous
version. In the version of WRF-Chem used in this study, it is separated to estimate
emission factor at each grid cell for isoprene and other BVOCs, respectively, although
the public offline MEGANv2.0 has the option to calculate isoprene emission factor
based on PFT. In WRF-Chem, the emission factor is prescribed for isoprene emission
at each grid cell, and calculated for other BVOCs using PFT-specified emission factors
and PFT cover database. The vegetation distributions can also be customized. For
activity factor, the impacts of PPFD, temperature, monthly LAI, leaf age, soil moisture,



and solar radiation on biogenic emissions are taken into account (Guenther, 2006).
However, there are some shortcomings in MEGANv2.0 implemented in public version
of WRF-Chem. MEGANv2.0 uses the monthly mean surface air temperature, LAI and
solar radiation from the climatological database that may not be consistent with the
meteorological fields during the simulation. In addition, the vegetation distribution at
each grid cell used in MEGANv2.0 (only 4 dominant PFT) is prescribed as different
from the one used in the land surface scheme (e.g., 24 land use types). Zhao et al. (2016)
implemented MEGANv2.1 into CLM4.0 in WRF-Chem so that the biogenic emission
scheme and the land surface scheme can use the consistent distributions of vegetation
type, surface air temperature, LAI and solar radiation. For emission factors,
MEGANv2.1 defines as the net primary emission that escaped into the atmosphere and
it does not contain the downward flux of chemicals from above canopy, while MEGAN
2.0 defines as the total flux of chemical compounds, detailed in Zhao et al. (2016).
Recently, Jiang et al. (2018) established the relationship between photosynthesis
and water stress and the BVOCs emissions that is not included in MEGANv2.1 in
WRF-Chem. They presented a more sophisticated mechanistic representation of
BVOCs emission in MEGAN (referred to as MEGANv3.0 hereafter) to simulate the
impact of drought on biogenic isoprene emissions. Following Jiang et al. (2018), the
MEGANv2.1 in WRF-Chem is updated to MEGANv3.0 in this study to include the
effect of drought on biogenic emissions, in which the new drought activity factor
$\gamma_{d,isoprene}$ is calculated as the following formula:
$\gamma_{d,isoprene} = 1 \qquad (\beta_t > 0.6)$
$\gamma_{d,isoprene} = V_{cmax}/\alpha \quad (\beta_t < 0.6, \alpha = 37)$ \hfill (2)
where $\alpha$ is an empirical and regionally applicable value derived from field
measurements at observation site in Missouri Ozarks AmeriFlux site (MOFLUX) to
limit and modify the isoprene emission due to the drought force. Therefore, the value
of $\alpha$ may not be suitable for China. However, due to the lack of observations in China,
the default $\alpha$ value is used in this study. $V_{cmax}$ is the photosynthetic enzyme activity, and
$\beta_t$ is the soil water stress function calculated as following:
$\beta_t = \sum w_i r_i$ \hfill (3)





where $w_i$ is the wilting factor based on soil water potential at each soil layer, and $r_i$
is the fraction of roots in soil layer. More details can be found in the CLM4.0 technical
notes (Oleson et al., 2010).

2.3 Vegetation distribution
Zhao et al. (2016) suggested that the distributions of vegetation types play an
important role in determining regional emissions of BVOCs with MEGAN. Two
vegetation datasets are used to examine the sensitivities of BVOCs emissions with
different MEGAN versions to vegetation distributions. One is the default land cover
dataset (USGS24) used in WRF-Chem (referred to as VEG-USGS hereafter), which
generally represents the land cover information for 1990s over East China (Loveland et
al., 2000). It is converted to 16-PFT data set in CLM4.0 (referred to as VEG-USGS
hereafter) following the table derived by Bonan (1996) as Zhao et al. (2016). Specific
descriptions of legend and class of the land cover data are listed in the Table 1. Another
land cover dataset is derived from the MODIS retrievals in 2015 (referred to as VEG-
2015 hereafter), which has the horizontal resolution of 1 km over entire China. VEG-
2015 were reclassified on the existing products of 2015, including GFSAD1000
(Cropland Extent 1km Crop Dominance, Global Food-Support Analysis Data)
(Thenkabail et al., 2012), and MODIS MCD12Q1 (MODIS Land Cover Type Yearly
Global 500m) product (Friedl et al., 2002). For MCD12Q1 product, there are six
different classification schemes (Gregorio, 2005), in which the two schemes of FAO
(Food and Agriculture Organization) LCCS (Land Cover Classification System) land
cover and FAO LCCS surface hydrology were used.
Figure 1 shows the spatial distributions of the dominant PFT within each model
grid cell (see details in Section 2.4) over East China from these two vegetation datasets.
It is apparent that VEG-2015 is much different from VEG-USGS. The Z-shaped urban
belt of Yangtze River Delta region is evident in VEG-2015 but not in VEG-USGS. Not
only the dominant PFT but also the sub-grid distributions of PFTs are different between
the two datasets (not shown). Table 1 illustrates the percentage of each PFT averaged
over the simulated domain from the two vegetation data sets. For example, the fraction



of needleleaf evergreen tree that is a major species of biogenic emission range from 7.9%
in VEG-USGS to 1.7% in VEG-2015, and the fraction of bare soil is nearly twice that
of VEG-USGS. The emissions of BVOCs from MEGAN could be significantly
different due to this difference. The sensitivity of MEGAN estimated BVOCs emission
to different vegetation distributions may also be different for different versions.

2.4 Numerical experiments

In this study, the simulations are conducted with a horizontal resolution of 12km

and 120×100 grid cells (109.3°E~125.6°E,25.4°N~36.4°N) over East China. The
simulation periods are April and July of 2015 representing one month of spring and
summer, respectively, to reflect the seasonal variation of biogenic emission. The quasi-
global WRF-Chem simulation with 360×145 grid cells (180°W~180°E,67.5°S~77.5°N)
at the 1°×1° horizontal resolution is used to provide the chemical boundary condition.
The meteorological initial and lateral boundary conditions are obtained from the NCEP
Final reanalysis data with 1°×1° resolution and updated every 6 hours. The modeled u
and v component wind and temperature in atmosphere above the planetary boundary
layer are nudged towards the NCEP Final reanalysis data with a 6-hour timescale
(Stauffer and Seaman, 1990).

Anthropogenic emissions for these simulations are obtained from the Hemispheric

Transport of Air Pollution verison-2 (HTAPv2) at 0.1°×0.1° horizontal resolution and
monthly temporal resolution for 2010 (Janssens-Maenhout et al., 2015), while the
Multi-resolutions Emission Inventory for China (MEIC) at 0.1°×0.1° horizontal
resolution for 2015 (Li et al., 2017b; Li et al., 2017c) is used to replace the emissions
over China within the simulation domain. Biomass burning emissions are obtained from
the Fire Inventory from NCAR (FINN) at 1 km horizontal resolution and hourly
temporal resolution (Wiedinmyer et al., 2011) and follow the injection heights proposed
by Dentener et al. (2006) in the Aerosol Comparison between Observations and Models
(AeroCom) and the diurnal variation provided by WRAP (2005). The GOCART dust
emission scheme (Ginoux et al., 2001) is used to calculated the vertical dust flux, and
the dust particles emitted into atmosphere are distributed by the MOSAIC aerosol size
bins based on the physics of scale-invariant fragmentation of brittle materials provided
by Kok (2011) . Sea-salt emissions is similar to Zhao et al. (2013a), which corrected
particles with radius less than 0.2 μm and considered the dependence of the temperature
of sea surface. More detailed about the sea-salt emissions and dust emission scheme
coupled with MOSAIC aerosol scheme in WRF-Chem can be found in (Zhao et al.,

2010)).

In order to investigate the sensitivities of simulated biogenic isoprene emissions
by different versions of MEGAN to different vegetation distributions, as mentioned
above, multiple experiments are conducted with different vegetation datasets and
MEGAN versions, as summarized in Table 2. First of all, three experiments are
conducted with the USGS vegetation distribution (VEG-USGS) using different
versions of MEGAN embedded in WRF-Chem as discussed above, i.e., MEGANv1.0
(Mv1-USGS), MEGANv2.0 (Mv2-USGS), and MEGANv3.0 (Mv3-USGS). The
sensitivities of biogenic emissions to different versions of MEGAN can be explored by
comparing these three experiments. Second, another three experiments are conducted
similar to the former ones but the VEG-USGS dataset is replaced by the VEG-2015
dataset, i.e., Mv1-2015, Mv2-2015, and Mv3-2015, respectively. By comparing these
two sets of experiments, the impacts of the two vegetation distributions on the simulated
BVOC emissions with each version of MEGAN can be investigated. These six
experiments are conducted for both April and July. The seasonal variation of the
sensitivities of BVOC emissions to different MEGAN versions and vegetation
distributions can be explored through the simulations for these two months.

## 3. Results

### 3.1 Biogenic isoprene emission

### 3.1.1 Sensitivity to emission schemes and vegetation distributions

Figure 3 shows the spatial distributions of biogenic isoprene emission averaged in
April for six simulations with different vegetation datasets and biogenic emission



schemes. First of all, with the same vegetation dataset of USGS, the large difference
exists among the results from these three versions of emission scheme. In terms of
domain average, MEGANv2.0 simulates the highest isoprene emission among the three
versions, MEGANv3.0 follows, and MEGANv1.0 simulates the lowest, especially over
the northwest of the simulation domain. It can also be noticed that the spatial
distributions of biogenic isoprene emission are different among the versions. To
illustrate better the difference, two focused areas (denoted by the red and black boxes
in Fig. 3) in the simulation domain are selected for further analysis. Over the southwest
region of domain (denoted by the black box), the averaged biogenic isoprene emission
in Mv1-USGS is below 0.2 mole/km$^2$/hr, and it is about 1.0 mole/ km$^2$/hr and 3.1
mole/km$^2$/hr from the Mv3-USGS and Mv2-USGS simulations, respectively. Over the
southeast region of domain (denoted by the red box), similarly, the MEGANv2.0
simulates the highest biogenic isoprene emission among the three versions and
MEGAN v1.0 estimates more emissions than MEGAN v3.0.

Over the southwest region of domain, for MEGAN v1.0, irrigated cropland (the

3rd land use type in VEG-USGS), cropland with grassland mosaic (the 5th), and
savanna (the 10th) are the dominant land use types over the southwest region (Fig. 1),
which have low emission factors (as shown in Fig. 2). For MEGAN v3.0, crop and grass
are the dominant PFTs over the region, but some temperate needle-leaf evergreen trees
that have higher emission factor of about 3 mg isoprene/m$^2$/hr (as shown in Fig. 3) are
also included in this area (Fig. 1). The different vegetation distributions lead to the
overall emission factors are different between MEGANv1.0 and MEGANv2.0 (Fig. 4).
Therefore, MEGAN v3.0 simulated more biogenic isoprene emissions than MEGAN
v1.0 (0.88 mole/km$^2$/hr versus 0.42 mole/km$^2$/hr) over this region. Over the southeast
region of domain, the dominant land use type is cropland with woodland mosaic (the
6$^{th}$) that has high emission factor of about 2 mg isoprene/m$^2$/hr and irrigated cropland
in MEGAN v1.0. By contrast, the PFTs in MEGAN v3.0 is crop and has lower emission
factor. This leads to larger overall emission factor in MEGANv1.0 than in MEGANv3.0
over this region. Therefore, MEGANv1.0 calculates more biogenic isoprene emissions
than MEGANv3.0 (1.08 mole/km$^2$/hr versus 0.65 mole/km$^2$/hr) over the area. In



general, the difference between MEGANv1.0 and MEGANv3.0 with the same USGS
land-use dataset is mainly due to the conversion of the USGS land-use to PFT that leads
to different vegetation types with different emission factors in each grid. For
MEGANv2.0, the emission factor of isoprene is obtained from the input database
directly in WRF-Chem, and it is the highest among the three versions of MEGAN (Fig.
4). Therefore, MEGANv2.0 simulates the most biogenic isoprene emissions over the
two analyzed regions among the three different versions independent of the vegetation
coverage (will be discussed below).

In terms of the modeling sensitivities to vegetation distributions (i.e., VEG-

USGS versus VEG-2015), as discussed above, with prescribed emission factor of
isoprene at each grid cell, the isoprene emission from MEGANv2.0 in WRF-Chem does
not change much with different vegetation distributions except some small perturbation
due to the impacts of vegetation distributions on meteorological fields. Over the
southwest of domain, the averaged biogenic isoprene emission with VEG-2015 is
higher (0.68 mole/km$^2$/hr and 2.25 mole/km$^2$/hr ) than that (0.42 mole/km$^2$/hr and 0.88
mole/km$^2$/hr) with VEG-USGS for both MEGANv1.0 and MEGANv3.0 due to the
increased fraction of needle-leaf evergreen tree and mixed forest over this area (Fig. 1)
in VEG-2015, and these land use types have higher emission factors (Fig. 2) than
croplands in VEG-USGS. Over the southeast, the vegetation coverage is significantly
reduced from VEG-2015 to VEG-USGS due to the rapid development in economic and
unban expansion over the region in last two decades. Therefore, for MEGANv1.0, the
averaged isoprene emission from Mv1-2015 is lower (0.39 mole/km$^2$/hr) than that (1.08
mole/km$^2$/hr) from Mv1-USGS, consistent with the lower overall emission factor with
VEG-2015 compared to VEG-USGS in MEGANv1.0 (Fig. 4). However, it is
noteworthy that, for MEGANv3.0, the isoprene emission from Mv3-2015 is higher
(1.12 mole/km$^2$/hr) than that (0.65 mole/km$^2$/hr) from Mv3-USGS. The different
sensitivities of the two versions to the vegetation changes are mainly due to their
different treatments of sub-grid vegetation distribution as discussed in Sect. 2.2, i.e.,
MEGANv3.0 considers sub-grid vegetation distribution besides the dominant
vegetation type at each grid cell when estimate the BVOCs emissions, while





MEGANv1.0 only considers the dominant vegetation type at each grid cell.

To further demonstrate the impact of sub-grid distribution of vegetation in

MEGANv3.0, Figure 5 shows the difference of major sub-grid fraction of PFT other
than the dominant one over the southeast region of domain within the red box between
VEG-2015 and VEG-USGS. Although the dominant vegetation types are crops, grass,
and bare soil over the region (Fig. 1), the sub-grid fractions of needle-leaf evergreen
tree, broad-leaf evergreen tree, and broad-leaf deciduous tree that have relatively higher
emission factors (Fig. 2) are higher in VEG-2015 than in VEG-USGS. As shown in Fig.
4, the overall emission factor $\varepsilon_{i,j}$ weighted by sub-grid PFT fractions is higher with
VEG-2015 than with VEG-USGS from MEGANv3.0. It highlights that the sub-grid
vegetation distribution is important in terms of estimating BVOC emissions over this
region, which results in more biogenic isoprene emission in MEGANv3.0 than
MEGANv1.0 with the latest vegetation distribution dataset (i.e., VEG-2015).

### 3.1.2 Environmental impacts

Besides emission factor, biogenic emission is also influenced by activity factor

that is largely controlled by environmental conditions. The activity factor mainly
accounts for the response of biogenic emission to temperature, leaf age, soil moisture,
solar radiation, leaf area index, and drought in current versions of MEGAN. The
seasonal variations (July versus April) of simulated biogenic emissions by different
versions of MEGAN with VEG-2015 are investigated to demonstrate the environmental
impacts and their difference among MEGAN versions. Please note that emission factors
are not dependent on seasons. Figure 6 shows the ratios of monthly averaged biogenic
isoprene emission and overall activity factor ($\gamma_i$) between July and April from three
versions of MEGAN with the VEG-2015 vegetation dataset. Only the simulation results
with VEG-2015 are analyzed here due to the similar results with VEG-USGS (not
shown). It is evident that the ratios are all greater than one, which means much more
isoprene is emitted into atmosphere by plant in July than in April. The seasonal
variations of magnitudes and the distributions of activity factors are consistent with
those of emissions. Among different versions, MEGANv3.0 is most sensitive to



environmental conditions, especially over the north of domain between 32°N and 36°N.
The overall activity factor is the product of the factors determined by temperature,
LAI, solar radiation, leaf age, and drought condition in MEGANv2.0 and MEGANv3.0
while it is an overall function of temperature and solar radiation in MEGANv1.0 that
cannot be separated as isolated factors. Therefore, for MEGAN v2.0 and MEGAN v3.0,
the ratios of isolated activity factors responding to temperature, LAI, solar radiation,
leaf age, and drought between July and April are further illustrated in Figure 7. The
temperature-dependent activity factor $(\gamma_t)$ plays an important role in the seasonal
change of total activity factor, especially in the north of simulation domain, which is
about 3.0~4.0 and >4.0 in MEGANv2.0 and MEGANv3.0, respectively, which means
that MEGAN predicts higher biogenic isoprene emission in warmer environment.
Guenther (2006) also point that the temperature-dependent activity factor increases
evidently with temperature.
There are two activity factors associated with leaves. One is related to the emission
dependence of absolute values of LAI $(\gamma_{LAI})$, which has the most different
distributions among all activity factors in two versions. The ratio in MEGAN v3.0 is
above 4 and < 2 in MEGAN v2.0 over the most part of simulation domain. It is
noteworthy that the ratio is below 1 in MEGAN v2.0 in the north of the domain while
it is more than 4 in MEGAN v3.0, that dominant the difference between July and April.
Figure 8 shows the change of activity factor for LAI as a function of LAI value used in
MEGAN v2.0 and MEGAN v3.0, respectively. It is evident that the estimated LAI
activity factors in both versions increase with the LAI values, but with different
increasing rates. In general, MEGANv3.0 has the faster increasing rate. Please note, as
discussed previously in Sect. 2.2, MEGAN v3.0 obtains the LAI online from the land
surface scheme directly that can capture the seasonal change well, while MEGAN v2.0
obtains it from climatological monthly mean input dataset that is different from the one
used in the land surface scheme in WRF-Chem. Figure 9 shows the distributions of LAI
in MEGAN v2.0 and MEGAN v3.0 in different months. For MEGAN v2.0, LAI has
almost no change from April to July, particularly over the northern simulation domain,
while for MEGAN v3.0, the LAI increases evidently over the whole domain, especially





over the north. Therefore, the ratio of $\gamma_{LAI}$ between July and April is around one in
MEGANv2.0, while it is much larger than one in MEGANv3.0.
The second is related to the leaf age $(\gamma_a)$ that also has quite different distributions
between MEGANv2.0 and MEGANv3.0. For MEGAN v2.0, the ratio is about 1~3 in
the most area while it below 1.0 between 30°N and 34°N, and more than 2 in the
northwest of simulation domain. For MEGAN v3.0, the ratio is above 1.0 over the
whole domain and the distribution has significant regional difference. It is 1~1.3 in the
south region while more than 2 in the north region. Generally speaking, leaf's ability to
emit biogenic isoprene is significantly influenced by leaf phenology. Young leaves emit
almost no isoprene, mature leaves emit mostly, and old leaves lose ability to produce
biogenic isoprene eventually. Therefore, plants emit more isoprene into atmosphere in
July than in April because of more mature leaves due to the plant growth. Activity factor
for leaf age is a function of the relative change of emission activity and the fraction of
leaves at different phenological stages that are determined by the difference of LAI in
current and previous month. In both versions of MEGAN, mature and old foliage have
highest relative isoprene emission activity, following by the growing foliage and the
new foliage is the lowest. Therefore, MEGAN v3.0 produce higher leaf age activity
factor in July because of the larger difference of LAI in different month and more
mature foliage to emit isoprene (Fig. 9).
The distribution of the ratio of light-depend activity factor $(\gamma_P)$ is also different
between the two versions. Light-depend activity factor $(\gamma_P)$ is a function of PPFD and
activity of isoprene synthase, and is dominated by the variation of PPFD. In general,
plants often have higher light-depend activity factor in July than that in April due to the
stronger radiation. For MEGAN v2.0, it is about 1~1.5 over the whole domain and has
no significant regional difference. For MEGAN v3.0, the ratio is below 1 in the south
of the domain while it is about 1.3~1.5 or more than 2 in the north of the domain.
MEGAN v3.0 considered the difference of sunlit and shaded leaves and PPFD will be
low on shaded leaves in dense canopy because of the blocking sunlight. Therefore,
MEGAN v3.0 calculated low light-depend activity factor in the south of the domain
due to the distribution of mixed forest which has dense canopy in summer.
The seasonal variation of drought-dependent activity factor $(\gamma_{d,isoprene})$ is only
included in MEGANv3.0 with the ratios of 1~3 over the domain. Previous studies have
shown that plants emit more isoprene into atmosphere under short-term mild drought
stress (e.g., Jiang et al., 2018). The reduction of stomatal conductance is accompanied
with the increase in leaf temperature resulting in the more isoprene emissions from
plants (Jiang et al., 2018). In addition, as discussed in the Section 2.2, drought-
dependent activity factor is proportional to photosynthetic enzyme activity, which can
be affected by PPFD. Therefore, MEGAN v3.0 estimated more isoprene emissions in
July especially in the north of domain and the pattern is similar to the distribution of
light-depend activity factor.

### 3.1.3 Comparison with observations

The results discussed above show the difference in modeling biogenic emissions
of isoprene. The difference of simulated near-surface isoprene concentrations is similar
as their emissions (not shown here). It will be optimal to compare the simulated
isoprene emissions and concentrations from different experiments with observations.
However, as far as we know, the publicly available in-situ measurements of isoprene
emissions and concentrations over East China is extremely sparse. Meanwhile,
although the satellite retrieved column integrated formaldehyde is often used for
evaluating simulated BVOCs, it is not suitable for using in this study because the
production from anthropogenic VOCs dominates the column formaldehyde over most
regions of East China. It is difficult to isolate the contribution from BVOCs to
formaldehyde from satellite retrievals. Since it is difficult to evaluate the simulated
results over a large area of East China, the limited observations are collected from
published literatures and unpublished data over both rural and urban areas of East China
to compare with the results of different experiments as listed in Table 3.
In general, the simulated results with MEGANv2.0 and MEGANv3.0 are closer to
measurements. MEGANv1.0 generally underestimates the observed values. Although
the results from MEGANv2.0 and MEGANv3.0 are quite different in some cases, they
are both within the uncertain range of observations in general. Although, as discussed
above, MEGANv3.0 can capture the biogenic isoprene emissions in urban area due to
its consideration of sub-grid vegetation distributions, it is difficult to be evaluated with
these limited data, except that at one site of Shanghai, the simulation with MEGANv3.0
produces higher surface concentration of isoprene that is closer to the observation
compared to that with MEGANv2.0. Overall, the experiments with MEGANv2.0 and
MEGANv3.0 may simulate better surface concentration of isoprene over East China
than that with MEGANv1.0, and more high-quality observations of BVOCs
concentrations in both rural and urban areas of East China are definitely needed to
further validate the modeling results in future.

**3.2 Impacts on mixing ratio of VOCs and ozone**

Difference in emissions of BVOCs from multiple versions of MEGAN can

influence the simulated mixing ratio of VOCs over East China that can further
significantly affect ozone production through photochemistry (Wei et al., 2007; Bao et
al., 2010; Calfapietra et al., 2013; Kim et al., 2013; Liu et al., 2018; Lu et al., 2019).
The photochemistry is most active in summer, therefore, the simulation results in July
with the latest vegetation coverage (VEG-2015) are analyzed here. Figure 10 shows the
distributions of total VOCs and HCHO concentrations near the surface contributed by
the BVOCs emissions simulated by the model with different versions of MEGAN using
VEG-2015 in July. It is evident that BVOCs contribute significantly to the amount of
total VOCs over East China, and the difference among the simulations with the three
versions of MEGAN is large. The simulation with MEGANv3.0 produces the highest
biogenic VOCs concentration (> 20 ppb), followed by MEGANv2.0 (10-20 ppb), and
the one with MEGAN v1.0 is the lowest (< 5 ppb), particularly over the northern region.
In terms of spatial distribution, the simulation with MEGANv3.0 generates higher
biogenic VOCs concentration over the north of domain, while the ones with the other
two versions of MEGAN generate higher concentration over the south, which is
consistent with the spatial distributions of the total biogenic emissions simulated by
different MEGAN versions in WRF-Chem (Fig. S1 in the supporting material). The
spatial distributions of simulated biogenic contribution to the surface formaldehyde





concentration are consistent with those of biogenic VOCs.
The significantly increased amounts of biogenic VOCs may induce the increase of
surface ozone concentration over East China (Zhao et al., 2009). Figure 11
demonstrates the spatial distribution of monthly mean ozone mixing ratio near the
surface contributed by the emissions of BVOCs. The simulation with MEGANv3.0
produces the largest amount of biogenic ozone over a large area of the simulation
domain. The biogenic ozone from the simulation with MEGANv3.0 is estimated over
8 ppb while it is 2~5 ppb from the one with MEGAN v2.0 and less than 1 ppb from the
one with MEGANv1.0. For MEGANv1.0 and MEGAN v3.0, the distributions of
surface biogenic ozone concentration is consistent with those of biogenic VOCs, for
example, MEGAN v3.0 estimated more biogenic VOCs over the north of the domain
while ozone concentration is also simulated higher in the same region. For MEGAN
v2.0, it is evident that the ozone formation is not influenced by biogenic VOCs solely.
The ozone production can be determined by the changes of both VOCs and NOx
concentrations, and the production efficiency can be different for $NO_x$-sensitive region
and VOCs-sensitive region (e.g., Zhao et al., 2009).
Figure 12 shows the surface concentrations of $NO_x$ due to the biogenic emissions
simulated with three versions of MEGAN with VEG-2015. The results are calculated
as the difference between simulations with and without biogenic emissions. The
simulations with MEGANv3.0 estimate the highest BVOCS-contributed concentration
change, especially over the north of domain (>2 ppb), followed by MEGAN v2.0 (0.2-
0.4ppb), and MEGAN v1.0 simulated lowest concentration (about 0.1ppb and below 0).
The different changes of surface NOx concentrations are mainly caused by the different
impacts on NOx lifetime due to biogenic VOCs. The increase of surface NOx
concentration is due to the BVOC-induced increase of NOx lifetime reflected by the
reduction of surface OH concentration (Fig. S2 in the supporting material). Therefore,
the increase of ozone contributed by biogenic emissions in the north of the domain
(30°N-36°N) simulated with MEGANv2.0 is due to the combined effect of increased
$NO_x$ and VOCs surface concentrations. It is also noteworthy that the surface ozone
concentrations are simulated lower over the southeast of domain than that in the


southwest with the three versions of MEGAN, while the surface concentrations of
BVOCs have no significant difference between the two regions. This is mainly due to
that the southwest is more sensitive to VOCs in terms of ozone production (Fig. S3 in
the supporting material) (e.g., Zhao et al., 2009).

**4. Summary and conclusion**
In this study, three versions of MEGAN in WRF-Chem and their difference in
simulating BVOC emissions and impacts on ozone mixing ratio over East China is
documented in the literature for the first time. The latest version of MEGAN v3.0 is
coupled within CLM4 land scheme as a part of WRF-Chem. Specifically, MEGAN v3.0
is updated from MEGAN v2.1 as an option in biogenic emission schemes and can share
the consistent vegetation map and other variables with CLM4 such as surface
temperature and leaf area index. What's more, MEGAN v3.0 include the activity factor
for drought and the combination of different versions of MEGAN and CLM4 are
employed to investigate the sensitivity of the variation of MEGAN versions.
Experiments are conducted for April and July over Eastern China with VEG-USGS and
VEG-2015 to study the sensitivities of simulated BVOCs by different MEGAN
versions in WRF-Chem to seasonal change and the distributions of vegetation. The
main conclusions are summarized below.
Physical and chemistry processes in these three versions of MEGAN implemented
in WRF-Chem are different, and the most intuitive distinction is their different
treatments of emission factor of BVOCs. MEGANv1.0 prescribed constant values for
different land use categories at each grid cell, and MEGANv2.0 has a stand-alone PFT
specific emission factor map. For MEGAN v3.0, the overall emission factor at each
grid cell is calculated by PFT-specific emission factor and the fraction of each PFT.
Therefore, the biogenic isoprene emissions estimated by three versions of MEGAN are
different over the simulation domain. The VEG-USGS and VEG-2015 datasets present
quite different distributions of vegetation coverage, which also contributes to the
difference of emission factors among different versions. Different versions of MEGAN



show different sensitivities to the changes of vegetation distributions due to their
different treatments of vegetation fraction in estimating emission factors of BVOCs.
The results highlight the importance of considering sub-grid vegetation fraction in
estimating BVOCs emissions. MEGANv3.0 with sub-grid vegetation distribution
simulates higher BVOCs emissions over the urban area of the Yangtze River Delta
(YRD) region compared to MEGANv2.0 with only the dominant vegetation type at
each grid cell.
Activity factor plays an important role in determining the seasonal change of
BVOCs emissions. Simulations with different versions of MEGAN show different
seasonal variation of activity factors and thus BVOCs emissions. The results indicated
that overall activity factor in July is higher than the one in April in all versions of
MEGAN, and MEGAN v3.0 is most sensitive to the seasonal change especially in the
north of simulation domain. In general, among all activity factors, temperature-
dependent factor dominates the seasonal change of activity factor in all three versions
of MEGAN, while the different response to the LAI change determines the difference
among the three versions in seasonal variation of BVOC emissions. The additional
drought-dependent activity factor in MEGANv3.0 can result in a little higher BVOC
emission over East China in July than April due to the increasing photosynthetic
enzyme activity, i.e., plants emit more biogenic isoprene in July than that in April under
the short-term mild drought forcing. The overall drought impact on BVOC emissions
over East China is small as previous studies (e.g., Jiang et al., 2018).
Different BVOCs simulated with the three versions of MEGAN in WRF-Chem
lead to the large difference in ozone production. The simulation with MEGANv3.0
produces the highest BVOCs contributed ozone concentration (> 8 ppbv) over East
China among the three versions, followed by the simulations with MEGANv2.0 and
MEGAN v1.0. The difference of BVOCs contributed ozone among the simulations with
three versions of MEGAN is not only affected by the increased concentration of
BVOCs but also influenced by the changes of NOx concentration. The simulations with
different versions of MEGAN show different distributions of surface NOx
concentration due to the BVOCs-induced changes of NOx lifetime. The production





efficiency of surface ozone concentration over East China due to BVOCs also depends
on the regions as $NO_x$-sensitive or VOCs-sensitive regions.
This study highlights that the simulated emissions of BVOC over East China is
sensitive to vegetation coverage, which has also been found by previous studies (e.g.,
Klinger et al., 2002; Wang et al., 2007). However, this study further demonstrates that
the modeling sensitivity to vegetation coverage could be quite different depending on
the BVOC emission schemes. Some studies also showed that BVOC emissions can be
more than 50% higher in summer than in other seasons (e.g., Li et al., 2020), which
may be also sensitive to the formulas of emission activity factors in different emission
algorithms as discussed in this study. Large uncertainties in modeling BVOCs emission
over East China still exist as reflected by different versions of the scheme, consistent
with previous studies that found the off-line calculation with different versions of
MEGAN led to significantly different BVOC emissions over China. Although it is
evident that surface ozone concentration can be significantly influenced by BVOC
emissions over East China through affecting VOCs, OH, and NOx and the BVOC
impact is also region-sensitive as found in this and previous works (e.g., Geng et al.,
2011; Tie et al., 2013; Liu et al., 2018), this study highlights that the overall impact can
be quite sensitive to different algorithms in emission schemes.
Due to these large uncertainties in emission factor, activity factor, and vegetation
distribution in estimating BVOC emissions over East China, the observations of BVOC
species such isoprene and monoterpene over East China are urgently needed to evaluate
the model and then further quantify the impacts of BVOCs on ozone and organic
aerosols. In addition, direct measurements of biogenic emission fluxes and/or emission
factors and activity factors are needed to constrain different emission algorithm in
atmospheric models. Last not the least, the survey of more accurate and higher
resolution vegetation distribution based on in-situ investigation and satellite remote
sensing should be conducted to support the estimation of BVOC emission over East
China.




**Data availability**

The released version of WRF-Chem can be downloaded from http://www2.mmm.ucar.edu/wrf/users/download/get_source.html. The updated USTC version of WRF-Chem can be downloaded from http://aemol.ustc.edu.cn/product/list/ or contact chunzhao@ustc.edu.cn. Also, the code modifications will be incorporated the release version of WRF-Chem in future.

**Author contributions**

Mingshuai Zhang and Chun Zhao designed the experiments, conducted and analyzed the simulations. All authors contributed to the discussion and final version of the paper.

**Acknowledgements**

This research was supported by the Fundamental Research Funds for the Central Universities, and the National Natural Science Foundation of China (grant 41775146), the USTC Research Funds of the Double First-Class Initiative, and the Strategic Proiority Research Program of Chinese Academy of Sciences (grant XDB41000000). The study used the computing resources from the High-Performance Computing Center of University of Science and Technology of China (USTC) and the TH-2 of National Supercomputer Center in Guangzhou (NSCC-GZ).



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



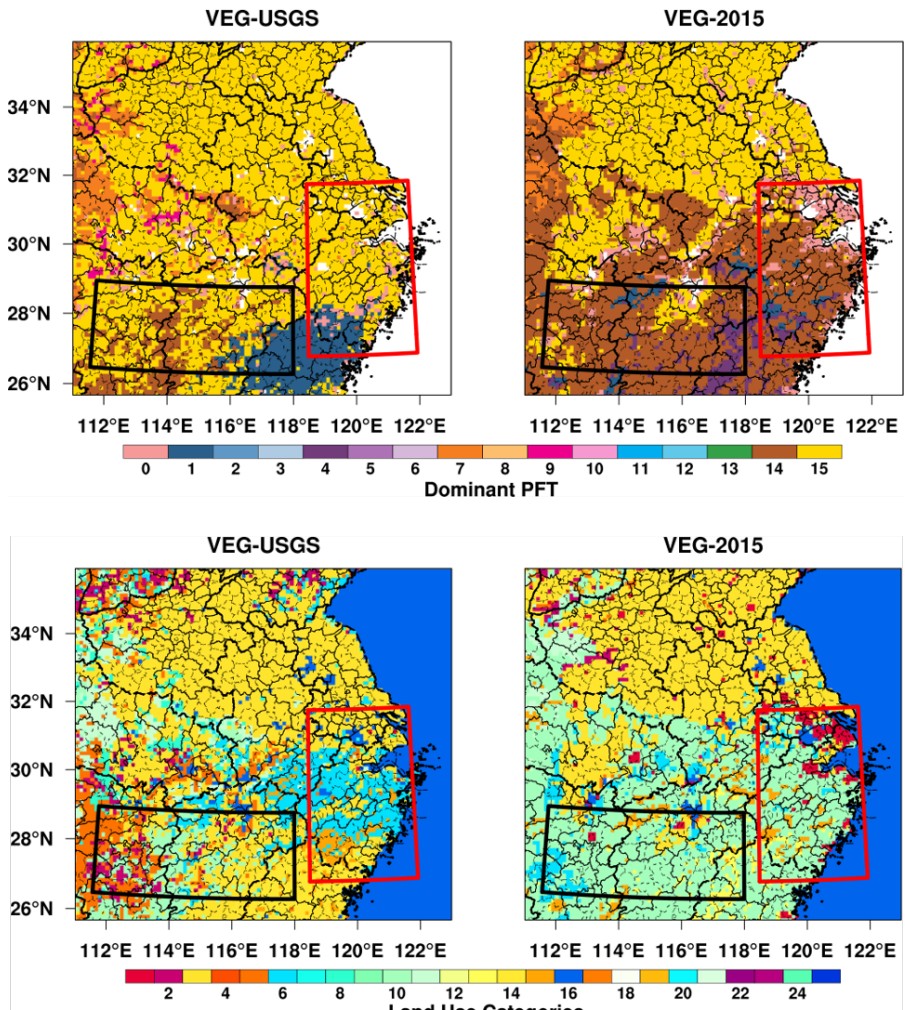

**Figure 1.** Spatial distribution of two different vegetation data sets (VEG-USGS and
VEG-2015) and dominant PFT derived from them in each grid over the simulation
domain.




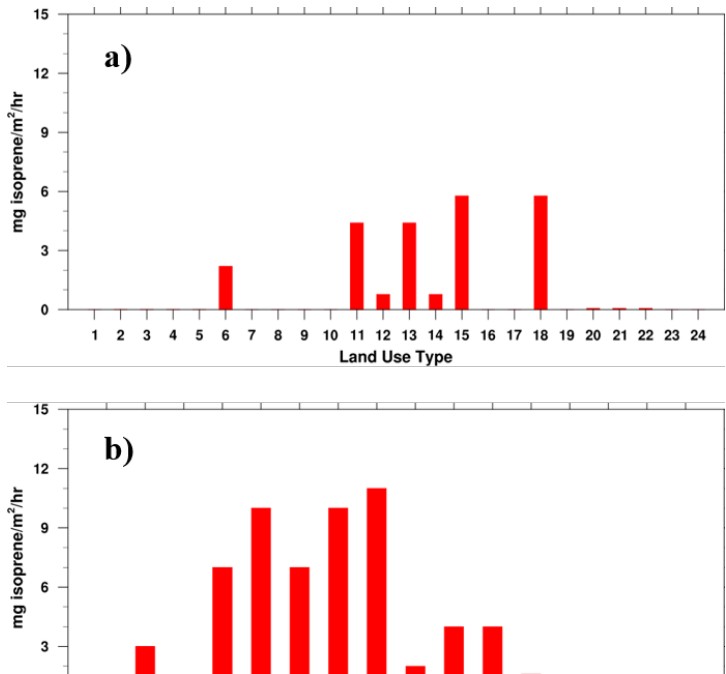


**Figure 2.** Biogenic emission factor for each land use category in **(a)** MEGAN v1.0,
and the land use number 1-24 detailed describe can be found in Table 3 and **(b)** for
each PFT in MEGAN v3.0, the PFT number 0-15 are listed in Table 1.




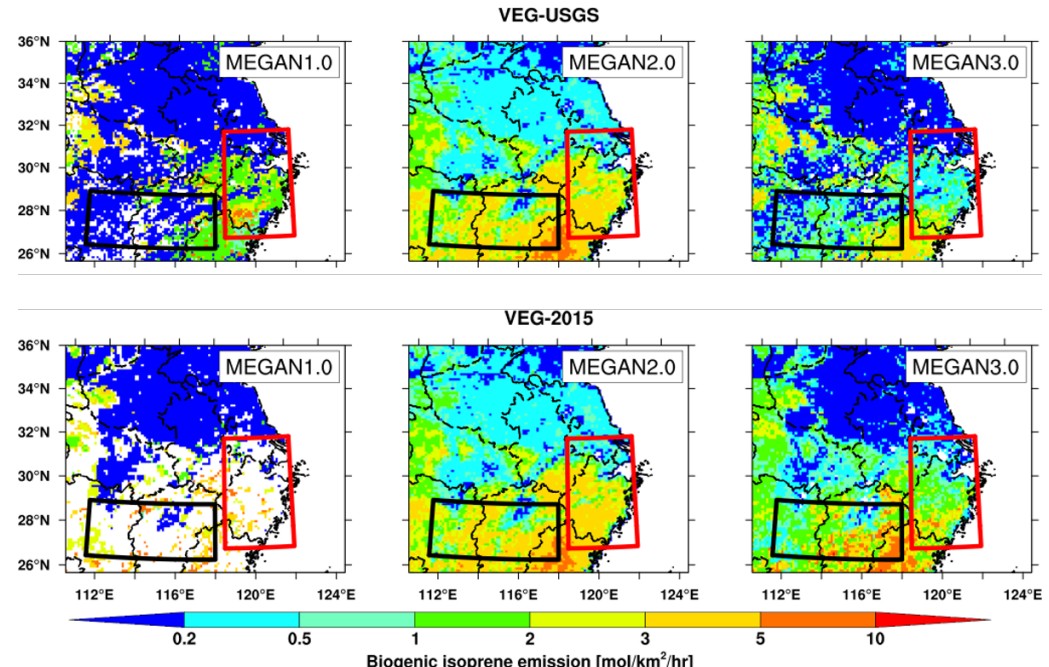


**Figure 3.** Spatial distributions of biogenic isoprene emissions averaged in April over the simulation domain estimated with different biogenic emission scheme and vegetation data set as listed in Table 1. Two areas are marked by red and black box to discuss the characters in detail.

















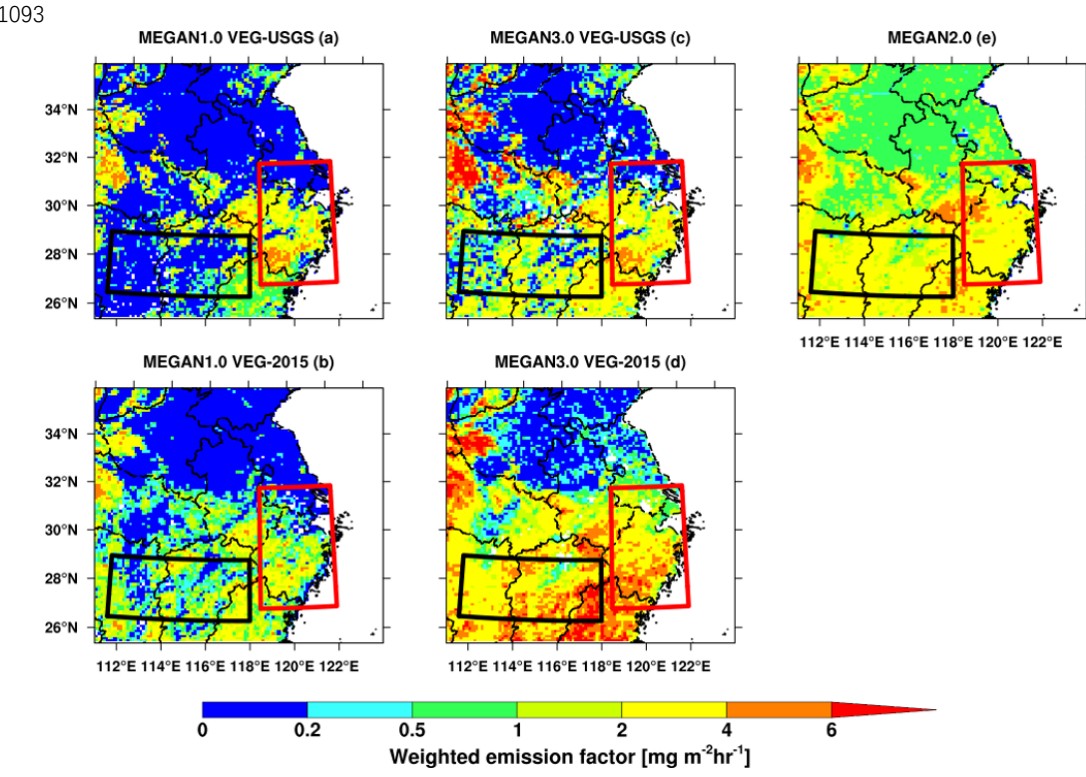


**Figure 4.** Spatial distribution of the weighted mean emission factor derived from VEG-USGS and VEG-2015 in MEGAN v1.0 **(a)(b)** and MEGAN v3.0 **(c)(d)**. Meanwhile, **(e)** shows the distribution of isoprene emission factor in MEGAN v2.0 database.






















**Figure 5.** Spatial distribution of the PFT percentage difference between the VEG-

2015 and VEG-USGS (VEG-2015 minus VEG-USGS) for needle-leaf evergreen tree,

broadleaf evergreen tree and broadleaf deciduous tree.














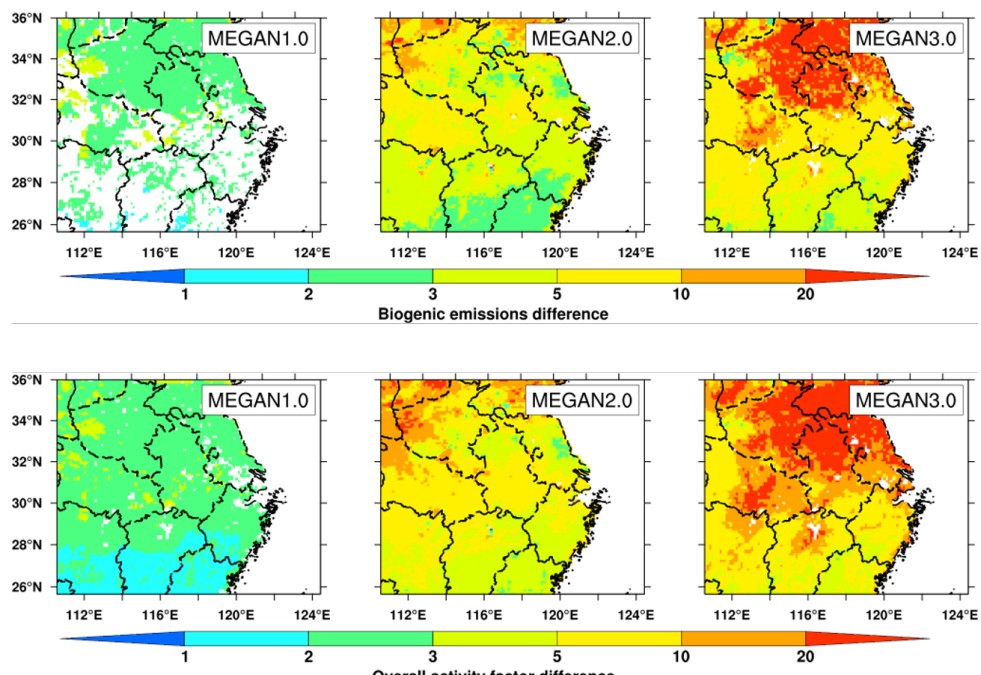

**Figure 6**. Spatial distribution of the quotient of biogenic isoprene emission and activity factor between simulations using VEG-2015 vegetation data set in July and that in April.



1144

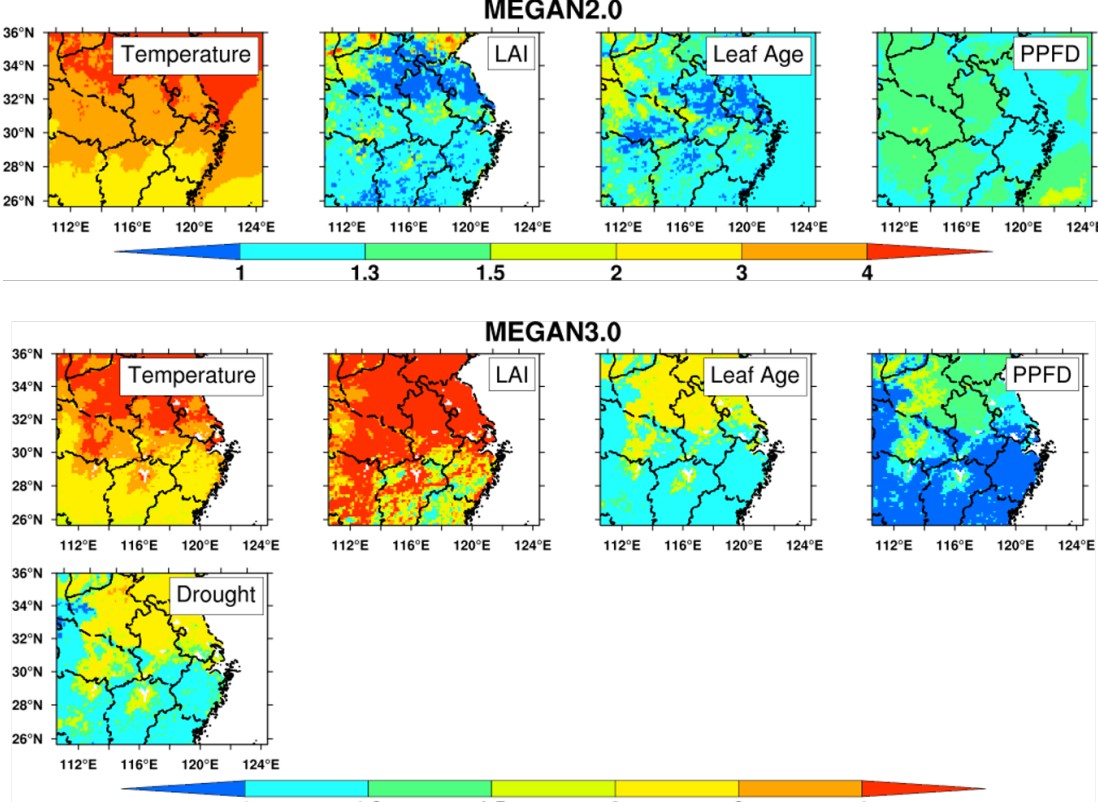

1145

**Figure 7**. Spatial distribution of the quotient of activity factor related to different environmental variables (such as temperature, LAI, light, leaf age and drought) between simulations in July and that in April (July divided by April) using VEG-2015 vegetation data set coupled with MEGAN v2.0 and MEGAN v3.0.
















*(figure plot showing γ_LAI vs LAI with MEGAN2.0 red line and MEGAN3.0 blue line)*



**Figure 8.** Activity factor for LAI $(\gamma_{LAI})$ variation with LAI in different versions of
MEGAN, red line represent the MEGAN v2.0 and blue line for MEGAN v3.0



















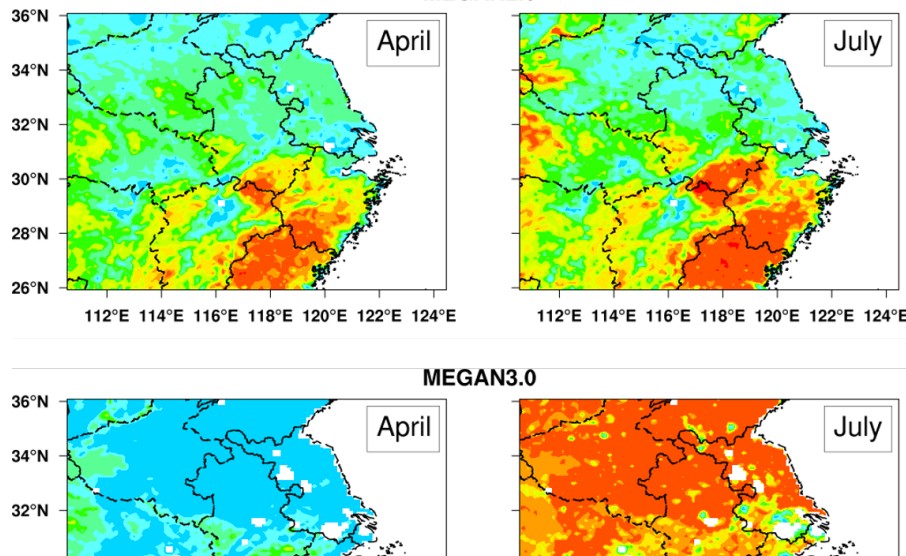



**Figure 9.** The spatial distribution of monthly leaf area index (LAI) from VEG-2015

in the MEGAN v2.0 and MEGAN v3.0.

















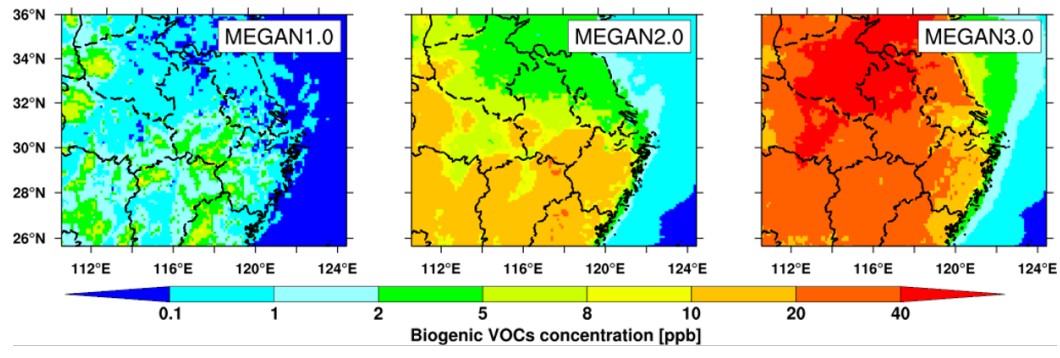

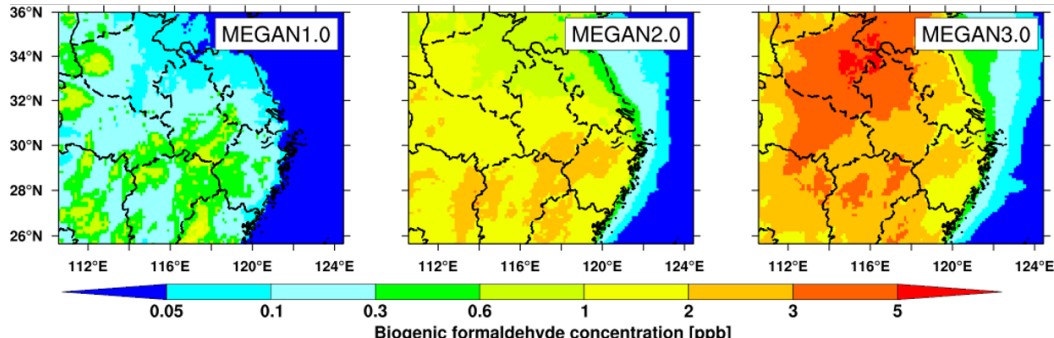


**Figure 10.** Spatial distribution of VOCs and formaldehyde concentration due to the

biogenic emissions (minus anthropogenic emissions) near the surface in July using

the VEG-2015 vegetation date set.


















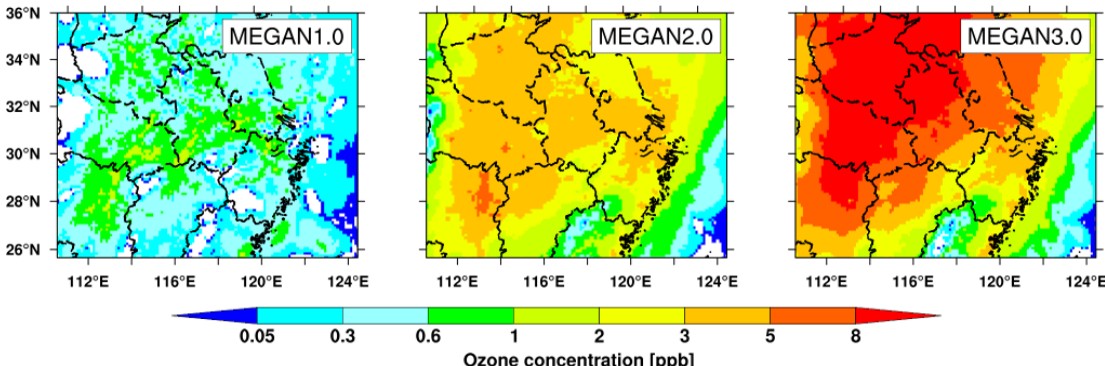


**Figure 11.** Spatial distribution of ozone concentration due to the biogenic emissions

near the surface in July using the VEG-2015 vegetation date set.






















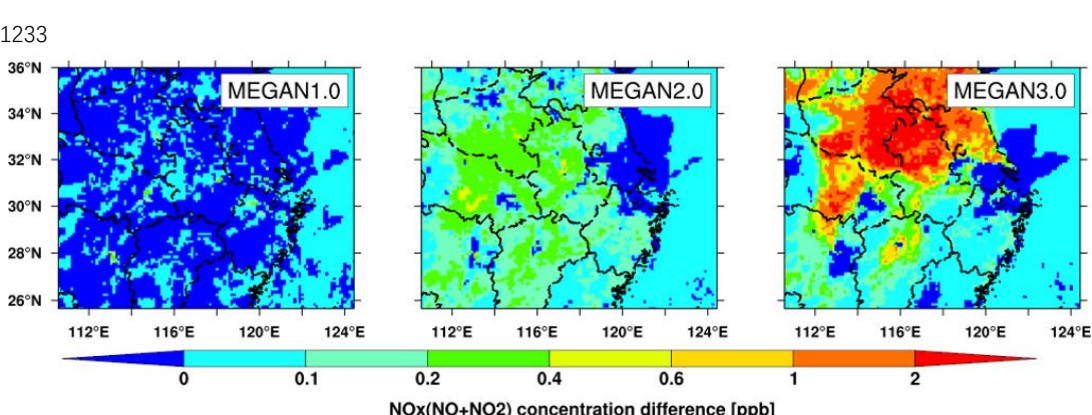


**Figure 12.** Spatial distribution of NOx concentration due to the biogenic emissions
near the surface (the difference of simulation considered biogenic emissions and the
one without biogenic emissions) in July using the VEG-2015 vegetation date set.





















**Table 1** The domain averaged fraction of PFTs in two vegetation data sets

| PFT number | description | VEG-USGS | VEG-2015 |
|:---:|:---:|:---:|:---:|
| 0 | Bare soil | 3.7 | 6.9 |
| 1 | Needleleaf evergreen tree–temperature | 7.9 | 1.7 |
| 2 | Needleleaf evergreen tree–boreal | 0 | 0 |
| 3 | Needleleaf deciduous tree–boreal | 0 | 0 |
| 4 | Broadleaf evergreen tree–tropical | 0 | 4.5 |
| 5 | Broadleaf evergreen tree–temperature | 0 | 0 |
| 6 | Broadleaf deciduous tree–tropical | 0 | 0 |
| 7 | Broadleaf deciduous tree–temperature | 7.6 | 4.4 |
| 8 | Broadleaf deciduous tree–boreal | 0 | 0 |
| 9 | Broadleaf evergreen shrub–temperature | 1.8 | 0 |
| 10 | Broadleaf deciduous shrub–temperature | 0 | 0 |
| 11 | Broadleaf deciduous shrub–boreal | 0 | 0 |
| 12 | $C_3$ arctic grass | 0 | 0 |
| 13 | $C_3$ grass | 0 | 0 |
| 14 | $C_4$ grass | 8.7 | 41.6 |
| 15 | Crop | 70.2 | 40.8 |








**Table 2** Numerical experiments of WRF-Chem in this study.

|  | Simulation period | BVOC scheme | Vegetation distribution | |
|---|---|---|---|---|
|  |  |  | VEG-USGS | VEG-2015 |
| WRF-Chem | April | MEGAN v1.0 | Mv1-USGS | Mv1-2015/Mv1-April |
|  |  | MEGAN v2.0 | Mv2-USGS | Mv2-2015/Mv2-April |
|  |  | MEGAN v3.0 | Mv3-USGS | Mv3-2015/Mv3-April |
|  | July | MEGAN v1.0 | - | Mv1-July |
|  |  | MEGAN v2.0 | - | Mv2-July |
|  |  | MEGAN v3.0 | - | Mv3-July |


































**Table 3** Measured and simulated isoprene concentrations (ppbv) at sampling sites.

| Location | Observation | Simulation | | | Source |
|---|---|---|---|---|---|
| | | MEGANv1.0 | MEGANv2.0 | MEGANv3.0 | |
| Lishui District, Nanjing | 0.062 | 0.010 | 0.071 | 0.049 | observation |
| Xujiahui commercial center, Shanghai | 0.120 | 0.020 | 0.075 | 0.090 | Cai et al. (2010) |
| Northern suburb, Nanjing | 0.960±0.56 | 0.017 | 0.905 | 0.073 | Shao et al. (2016) |
| Nanjing University of Information Science&Technology | 0.300±0.35 | 0.023 | 0.171 | 0.180 | Li et al. (2014) |
| Hubei Provincial Environmental Monitoring Center, Wuhan | 0.380(0.18-0.6) | 0.130 | 0.997 | 1.090 | Lyu et al. (2016) |
