# Peer review of "Supporting materials for “Sensitivity of different BVOC emission schemes in WRF-Chem to vegetation distributions and its impacts over East China”"

_Geoscientific Model Development, 2021_

## Referee Comment (RC1)

**Sensitivity of different BVOC emission schemes in WRF-Chem (v3.6) to vegetation distributions and its impacts over East China**

**Abstract (Line – comment)**

41 - … the version of WRF-Chem updated… → You should use the acronym since it is the first time you mentioned WRF-Chem.

**1. Introduction (Line – comment)**

115 - … and therefore is mainly used in previous studies (e.g., (Guenther et al., 1996…. → I think the double parenthesis is a typo…

132 - production over East China (e.g., (Han et al., 2005; Wei et al., 2007; → as line 115

**General comment on abstract and introduction**

The abstract and introduction have a well-defined and clear structure, but for few mistakes along the way, they look fine.

**2. Methodology (Line – comment)**

171-182 – I think you could use a table to summarize the configuration of WRF-Chem.

191 – Since you introduced the update made in MEGAN v2.1 by Zhao et al. 2016, I think you can move the lines 224-230 (from Zhao et al to Zhao et al…) here.

259-260 – the sentence "(referred to as VEG-USGS hereafter)" is a repetition, you could delete it

262 – The acronym of MODIS is missing

285-295 – A map of the simulation domain could be clarifying the visualization of it to the readers.

296-313 – As the WRF-Chem configuration you can use a table summarizing the chemical mechanism and emissions used, maybe you could use one table for the whole WRF-Chem setup.

313 – Typo in the citation Zhao et al.

**General comment on the methodology**

The methodology paragraph is well-structured as the introduction, but I think you should go through the paragraphs and represent more efficiently the simulation domain and (physic and chemistry) parametrization by a table and/or map.

**3. Results (Line – comment)**

357 – I think the reference should be to Fig. 3 instead of Fig. 4

386 - The word "unban" is a typo.

Fig. 5 – If you write on the caption the number of PFT for needle-leaf evergreen tree, broadleaf evergreen tree and broadleaf deciduous tree, it could be clearer to the readers.

Fig. 6 – It could be better to explain the ration in the caption, as you did in the Fig. 7 (i.e., "…between simulations in July and that in April…")

Fig. 8 – The plot misses of the axis values.

463-472 – I think you should introduce here the Guenther et al., 2006 citation.

**4. Summary and conclusion** (Line – comment)
No comments

**General Comment on results and conclusion**
1) I very appreciated as you presented the results and I believe it is a very useful study since you clarified some of the main changes between the different version of MEGAN models.
2) The study lacks an exhaustive comparison with the observed data of BVOC, I appreciate your commitment in reporting data from the literature to get a general idea of the comparison, but I think that this aspect should be deepened, perhaps with the next study.

---

## Author Comment (AC2)

**Mauro Morichetti Referee #1**

*Abstract(Line-comment)*

- *41 - … the version of WRF-Chem updated…→You should use the acronym since it is the first time you mentioned WRF-Chem.*

Thanks for correction. Now it is revised as "… the version of Weather Research and Forecasting model coupled with Chemistry (WRF-Chem) updated…"

*1. Introduction (Line – comment)*

- *115 - … and therefore is mainly used in previous studies (e.g., (Guenther et al., 1996…. → I think the double parenthesis is a typo…*

Corrected as suggestion.

- *132 - production over East China (e.g., (Han et al., 2005; Wei et al., 2007; → as line 115*

Corrected as suggestion.

- *General comment on abstract and introduction*
  *The abstract and introduction have a well-defined and clear structure, but for few mistakes along the way, they look fine.*

We thank the reviewer for the detailed and constructive comments. They are very helpful for improving the fluency and quality of the manuscript.

*2. Methodology (Line – comment)*

- *171-182 – I think you could use a table to summarize the configuration of WRF-Chem.*

Thanks for your suggestion. We add a table in the revised manuscript (Table 1) to summarize the detailed experiment configuration of this study.

- *191 – Since you introduced the update made in MEGAN v2.1 by Zhao et al. 2016, I think you can move the lines 224-230 (from Zhao et al to Zhao et al…) here.*

Thanks for your suggestion. Now we clarify the descriptions about three versions of MEGAN in the revised manuscript as "Three versions of MEGAN (MEGANv1.0, MEGANv2.0, and MEGANv3.0) online coupled with WRF-Chem are investigated in this study. The details about these three different versions are described below."

- *259-260 – the sentence "(referred to as VEG-USGS hereafter)" is a repetition, you could delete it.*

Corrected as suggestion.

- *262 – The acronym of MODIS is missing.*

Thanks for checking. Now it is revised as Moderate-resolution Imaging Spectroradiometer (MODIS).

- *285-295 – A map of the simulation domain could be clarifying the visualization of it to the readers.*

Thanks for your suggestion. Fig. S1 in the revised supporting material of manuscript shows the simulation domain, and we clarify it in the revised manuscript as "In this study, the simulations are conducted with a horizontal resolution of 12 km and $120 \times 100$ grid cells (109.3°E~125.6°E,25.4°N~36.4°N; Fig. S1 in the supporting material) over East China."

- *296-313 – As the WRF-Chem configuration you can use a table summarizing the chemical mechanism and emissions used, maybe you could use one table for the whole WRF-Chem setup.*

Thanks for your suggestion. Now Table 1 is added in the revised manuscript to summarize the WRF-Chem configuration.

- *313 – Typo in the citation Zhao et al.*

Corrected.

- *General comment on the methodology*
  *The methodology paragraph is well-structured as the introduction, but I think you should go through the paragraphs and represent more efficiently the simulation domain and (physic and chemistry) parametrization by a table and/or map.*

Thanks for your suggestion. The simulation domain and the Table for model configuration are added as our response to your comments above.

*3. Results (Line – comment)*
- *357 – I think the reference should be to Fig. 3 instead of Fig. 4.*

Thanks for checking. Corrected as Fig. 3.

- *386 - The word "unban" is a typo.*

Corrected as "urban".

- *Fig. 5 – If you write on the caption the number of PFT for needle-leaf evergreen tree, broadleaf evergreen tree and broadleaf deciduous tree, it could be clearer to the readers.*

Thanks for your suggestion. Now we add the PFT numbers for needle-leaf evergreen tree, broadleaf evergreen tree, and broadleaf deciduous tree in the caption of Fig. 5.

- *Fig. 6 – It could be better to explain the ration in the caption, as you did in the Fig. 7 (i.e., "...between simulations in July and that in April…")*

Thanks for your suggestion. Now the caption of Fig. 6 is revised as "**Figure 6**. Spatial distribution of the quotient of biogenic isoprene emission and activity factor between simulations in July and that in April, using VEG-2015 vegetation data set coupled with different emission schemes (MEGANv1.0, MEGANv2.0 and MEGANv3.0)."

- *Fig. 8 – The plot misses of the axis values.*

Thanks for your checking. We correct it in the Fig. 8 of revised manuscript.

- *463-472 – I think you should introduce here the Guenther et al., 2006 citation.*

Thanks for your suggestion. We agree with you and cite Guenther et al. 2006 here in the revised manuscript. Now the sentence is revised as "Generally speaking, leaf's ability to emit biogenic isoprene is significantly influenced by leaf phenology. Young leaves emit almost no isoprene, mature leaves emit mostly, and old leaves lose ability to produce biogenic isoprene eventually ( Guenther et al., 2006). Therefore, plants emit more isoprene into atmosphere in July than in April because of more mature leaves due to the plant growth. Activity factor for leaf age is a function of the relative change of emission activity and the fraction of leaves at different phenological stages that are determined by the difference of LAI in current and previous month, introduced by Jiang et al. (2018)."

*4. Summary and conclusion (Line – comment)*
*No comments*

*General Comment on results and conclusion*

- *1) I very appreciated as you presented the results and I believe it is a very useful study since you clarified some of the main changes between the different version of MEGAN models.*

We appreciate the reviewer for the valuable comments. Your detailed suggestions played a key role in more fluent expressions and more clear structures of this manuscript. We hope that the main difference among different versions of MEGAN in WRF-Chem discussed in this study can provide useful information to the WRF-Chem community using MEGAN.

- *2) The study lacks an exhaustive comparison with the observed data of BVOC, I appreciate your commitment in reporting data from the literature to get a general idea of the comparison, but I think that this aspect should be deepened, perhaps with the next study.*

Thanks for the reviewer's understanding. The public in-situ measurements of BVOCs emissions and concentrations over East China is extremely sparse so that we can only collect limited published observations from literature to evaluate the model.

In fact, we did think about using satellite retrieval of formaldehyde to indirectly evaluate the simulated BVOCs. However, we concerned about the difficulties to isolate the contributions from anthropogenic and biogenic BVOCs in retrievals, therefore we didn't show the comparison in the original manuscript. In order to respond to other reviewers' comment, now we evaluate the simulated total column tropospheric formaldehyde with satellite retrievals in April and July. We add Fig. 10 in the revised manuscript to show the monthly mean total column formaldehyde concentration simulated by different versions of MEGAN in April and July and compared with satellite retrievals. In general, the simulated tropospheric column formaldehyde concentrations are consistent with satellite retrievals in April, showing high column formaldehyde concentration over the Yangtze River Delta region and South China. Based on Fig. S7, the formaldehyde concentrations are contributed comparably by both anthropogenic and biogenic sources over these two regions in July, and biogenic source contributes about 20% to the total in April, particularly over the southern domain and with MEGANv2.0 and v3.0. Although there are some small differences in formaldehyde column concentrations in April among the simulations with different MEGAN versions, it is difficult to apply the satellite retrievals to constrain their small difference if considering the uncertainties of retrievals. In July, the difference between the simulations and satellite retrievals is larger than that in April. The difference among the simulations with different MEGAN versions is also much larger. Compared to the satellite retrievals, the simulation with MEGANv1.0 (MEGANv3.0) may underestimate (overestimate) tropospheric formaldehyde column concentrations. These biases may reflect their errors in biogenic emissions. The large difference between MEGANv2.0 and MEGANv3.0 in July may indicate that some activity factors controlling seasonal variation of BVOCs emissions is less appropriate in MEGANv3.0 than in MEGANv2.0. However, considering the uncertainties of satellite retrievals and anthropogenic VOC emissions, satellite retrieval of formaldehyde alone is still difficult to constrain uncertain parameters or functions in BVOCs emission scheme over the regions like China. More direct observations of BVOCs at multiple sites or from aircraft for different seasons are needed to evaluate overall model performance of BVOCs over a region, which will be achieved in future studies. More details about the comparison with satellite retrievals and discussion can be found in the revised manuscript.

**Anonymous Referee #2**

*General comments:*

- *This paper describes differences in BVOC emissions resulting from different versions of the MEGAN model, along with the effect on ozone and other trace gases over portions of China. Differences due to the specification of vegetation types are also examined. Parameterizations developed for specific processes in air quality models, such as online treatments of biogenic emissions, continually undergo modification as new information is available to constrain the relationships used by the parameterization. So, it is not surprising that the estimated emission rates are different among the versions of MEGAN.*

We thank the reviewer for the detailed and constructive comments. They are very helpful for improving the quality of the manuscript. We agree that it is always not surprising that there are differences in modeling results among different versions of parameterization. It is also expected that the update in the newer version can improve the model performance, which, however, is not always true as we all know. Therefore, it is necessary to document and demonstrate the difference among different versions because the modeling community never automatically move to the newer version in default. In fact, it is quite common to investigate the difference among different versions of parameterization in the modeling community (i.e., LeGrand et al., 2019; Zhao et al., 2019; Onwukwe and Jackson, 2020; Zeng et al., 2020).

Although the comparison among different versions were explored by Zhao et al. (2016) over California, they focused on emission factors and fluxes only and did not pay much attention on different activity factor functions among different versions. In addition, the WRF-Chem with MEGAN was widely used in China but was not documented well before. Therefore, this study aims to examine the difference among different versions of MEGAN used in WRF-Chem for East China, particularly between MEGANv2.0 and MEGANv3.0 because both of them are still widely used. The comparison of modeling results with different versions of MEGAN can help us understand the mechanisms leading to the difference. Furthermore, we also examine the major changes among different versions of MEGAN under seasonal variation and with different vegetation datasets. This analysis can find the most sensitive parameters (e.g., temperature and leaf area index) that lead to the major difference, which can help us to identify the key processes that needs more accurate observations to constrain. We thus consider our manuscript appropriate for Geoscientific Model Development (GMD).

Lastly but not least, although we currently don't have enough observations to evaluate the modeling results, the analysis of difference among different versions can provide useful information about what kind of observations are more urgently needed and over which region of China. For example, this study points out that the sensitivities of biogenic

emissions to LAI and temperature are large over China, and thus the laboratory and field experiments are needed to constrain better the functions.

To clarify the significance of this study, we add more clarification and discussion in the revised manuscript:
-In introduction:
"Since the WRF-Chem model with different MEGAN versions has been widely used for studying the impacts of BVOCs on air quality over East China while the performance of different MEGAN versions in WRF-Chem has not been examined, this study aims to investigated the difference of MEGAN versions in terms of modeling BVOCs, focusing on biogenic isoprene, and its impact on ozone concentration over East China."
"In summary, this study documents the different performance among different versions of MEGAN and its impacts on ozone and other chemical compounds, which can provide the WRF-Chem community more comprehensive analysis to understand the mechanisms of modeling sensitivities in BVOCs among different versions of MEGAN in WRF-Chem and vegetation distributions. The different response to seasonal change and vegetation distributions are also quantified. On the other hand, modeling sensitivity analysis also provides more information about what and where to measure for better constraining the modeling of BVOCs over East China."

-In methodology:
"However, different versions of MEGAN implement different treatments for calculating emission rates and environmental forcing, and therefore, the detailed difference of these versions of MEGAN in WRF-Chem and their impacts on modeling results needs to be quantified"
"The sensitivity of estimated BVOCs emissions to these two vegetation distributions may also be different due to the different treatment of vegetation type in three versions of MEGAN used in this study."

In results:
"In general, the difference between MEGANv1.0 and MEGANv3.0 with the same USGS land-use dataset is mainly due to the conversion of the USGS land-use to PFT that leads to different vegetation types with different emission factors in each grid. For MEGANv2.0, the emission factor of isoprene is obtained from the input database directly in WRF-Chem, and it is the highest among the three versions of MEGAN."
"It highlights that the sub-grid vegetation distribution is important in terms of estimating BVOC emissions over this region, which results in more biogenic isoprene emission in MEGANv3.0 than MEGANv1.0 with the latest vegetation distribution dataset (i.e., VEG-2015)."

-In summary and conclusion:

"In this study, three versions of MEGAN in WRF-Chem and their difference in simulating BVOC emissions and impacts on ozone mixing ratio over East China is documented in the literature for the first time. The latest version of MEGAN v3.0 is coupled within CLM4 land scheme as a part of WRF-Chem."

"Physical and chemical processes in these three versions of MEGAN implemented in WRF-Chem are different, and the most intuitive distinction is their different treatments of emission factor of BVOCs."

"This study highlights that the simulated emissions of BVOC over East China is sensitive to vegetation coverage, which has also been found by previous studies (e.g., Klinger et al., 2002; Wang et al., 2007). However, this study further demonstrates that the modeling sensitivity to vegetation coverage could be quite different depending on the BVOC emission schemes."

"High-quality direct observations of BVOCs emissions or concentrations for different season at multiple sites or from aircrafts in both rural and urban areas of East China are definitely needed to evaluate overall model performance of BVOCs over China, particularly over some specific areas with large modeling sensitivities of BVOC emission and activity factors, such as the Anhui and Henan provinces in the north of simulation domain, suggested by this study."

- ***The other conclusion is that specification of the subgrid vegetation fraction is also important, which was already shown in Zhao et al. (2016) (and noted in lines 253-254) who is a co-author on this paper.***

Although the influence of subgrid vegetation fraction in MEGANv2.1 was discussed in Zhao et al. (2016), its impacts on BVOC concentration simulated by different versions of MEGAN in WRF-Chem are not documented and quantified yet in China, which encounters the rapid urbanization in recent years. In this study, we replace the default vegetation distribution in WRF-Chem (representing the land-use information for 1990s) with a new land-use dataset reflecting the urbanization in China in 2010s, and show that the BVOCs concentration will be underestimated in urban area of China if the sub-gird distributions are ignored, which was not discussed in Zhao et al. (2016). What's more, the investigation of sub-grid impact is also not the focus of this study.

To clarify this, the sentence in abstract is revised as "In particular, the results highlight the importance of considering sub-grid vegetation fraction in estimating BVOCs emissions over East China with large area of urbanization."

*Major comments:*
- ***The authors conclude that their results show there is still a large uncertain range in modeling BVOCs (line 61); however, this begs the question: What is uncertainty?***

*In terms of the paper presented, the range of results are from different versions of MEGAN. This sort of model-to-model uncertainty might be even larger if one considers other treatments of BVOC emissions (listed in lines 88-91). But what is the value of using an older version of MEGAN when a newer version is available and has presumably been shown to perform better and/or have more physical processes represented than the older version?*

Thanks for your suggestion and raising an important point. As you mentioned, this study defines "uncertainty" as the modeling difference among different versions of MEGAN parameterization. This may be mis-leading in different communities. Therefore, we change "uncertainty" to "sensitivity" through the revised manuscript. The title of manuscript is also revised as "Modeling sensitivity of BVOC to different versions of MEGAN schemes in WRF-Chem(v3.6) and its impacts over East China".

As our response to your comments above, it is often expected that the update in the newer version can improve the model performance, which, however, is not always true as we all know. Therefore, it is necessary to document and demonstrate the difference among different versions because the modeling community never automatically move to the newer version in default. In fact, it is quite common to investigate the difference among different versions of parameterization in the modeling community (i.e., LeGrand et al.,2019; Zhao et al., 2019; Onwukwe and Jackson, 2020; Zeng et al., 2020).

This study aims to examine the difference among different versions of MEGAN used in WRF-Chem for East China, particularly between MEGANv2.0 and MEGANv3.0 because both of them are still widely used. The comparison of modeling results with different versions of MEGAN can help us understand the mechanisms leading to the difference. Furthermore, we also examine the major changes among different versions of MEGAN under seasonal variation and with different vegetation datasets. This analysis can find the most sensitive parameters (e.g., temperature and leaf area index) that lead to the major difference, which can help us to identify the key processes that needs more accurate observations to constrain.

We agree that the observations are needed to explore the uncertainties in modeling BVOCs, however, it is quite difficult to collect effective observations of BVOCs over China to evaluate the modeling results. Now, in the revised manuscript, we add some comparison with satellite retrieved formaldehyde. Please see our response to your other comments.

- *What is lacking here is a comparison of predicted BVOCs with observations to truly understand uncertainty. Maybe there are no observations of BVOCs such as isoprene or monoterpene in these regions for the simulation periods. Perhaps the simulations could have been done for periods when such observations are available*

*and/or for different parts of China is needed. I would assume that at a minimum some air quality data, such as ozone and NOx, would be available to compare with the model results to indirectly assess the effect of BVOC emissions. Section 3.1.3 seems to suggest there are some observations, but they chose not to show any results.*

It will be great if there are some observations for evaluation. However, the direct observations of BVOCs are really scarce over China. We collected some observations from published literatures and a commercial measurement (only in Lishui District, Nanjing) as listed in Table 4. Please note that this commercial measurement has not been published and is not under quality-control, therefore we decide not to discuss about it too much. Those observations were collected at different sites for different periods. Now we add the time information of observations in Table 4 in the revised manuscript. Ideally, we need the observations at multiple sites or from aircraft for a specific period to evaluate overall model performance of BVOCs over a region (e.g., Zhao et al., 2016). Therefore, it is difficult to evaluate effectively any simulations with those observations listed in Table 4.

Although we do have observations of ozone and NOx at multiple sites over East China, it is also difficult to evaluate the simulated BVOCs with those observations. First, NOx and ozone can be largely affected by NOx emission that is quite uncertain over East China. Therefore, it is hard to indirectly evaluate the simulated BVOCs with NOx and ozone. Second, although ozone can be largely affected by VOCs and can reflect the strength of VOCs to some extent, VOCs are contributed comparably by anthropogenic and biogenic VOC over East China, which is not like some regions such as the southeastern United States and the Amazon area where VOCs are dominated by BVOCs. Formaldehyde is often used to reflect the overall burden of VOCs. Fig. S7 shows the spatial distribution of anthropogenic and biogenic column formaldehyde concentration in April and July from the simulations with VEG-2015. It is evident that anthropogenic and biogenic formaldehyde concentrations are comparable over most regions of East China in July, and biogenic source contributes about 20% to the total in April. On the whole, it is difficult to use observations of ozone and NOx to evaluate the simulated BVOCs over East China.

In fact, we did think about using satellite retrieval of formaldehyde to indirectly evaluate the simulated BVOCs. However, as we discussed above, we concerned about the difficulties to isolate the contributions from anthropogenic and biogenic BVOCs in retrievals, therefore we didn't show the comparison in the original manuscript. Upon the reviewers' suggestion, now we evaluate the simulated total column tropospheric formaldehyde with satellite retrievals in April and July. We add Fig. 10 in the revised manuscript to show the monthly mean total column formaldehyde concentration simulated by different versions of MEGAN in April and July and compared with satellite retrievals. In general, the simulated tropospheric column formaldehyde concentrations are consistent with satellite retrievals in April, showing high column formaldehyde concentration over

the Yangtze River Delta region and South China. Based on Fig. 10 and Fig. S7, the formaldehyde concentrations are contributed comparably by both anthropogenic and biogenic sources over these two regions in July, and biogenic source contributes about 20% to the total in April. Although there are some small differences in formaldehyde column concentrations in April among the simulations with different MEGAN versions, consistent with the comparison of biogenic emissions (Fig. 3), it is difficult to apply the satellite retrievals to constrain their small difference if considering the uncertainties of retrievals. In July, the difference among the simulations with different MEGAN versions is much larger. Compared to the satellite retrievals, the simulation with MEGANv1.0 (MEGANv3.0) may underestimate (overestimate) tropospheric formaldehyde column concentrations. These biases may reflect their errors in biogenic emissions. The large difference between MEGANv2.0 and MEGANv3.0 in July may indicate that some activity factors controlling seasonal variation of BVOCs emissions is less appropriate in MEGANv3.0 than in MEGANv2.0. However, please note that satellite retrievals of formaldehyde may also have relatively large uncertainties in July (e.g., Su et al., 2019; Su et al., 2020) and the uncertainties of anthropogenic emissions of VOCs may also contribute to the modeling biases of formaldehyde. As we mentioned above, more direct observations of BVOCs at multiple sites or from aircraft for different seasons are needed to evaluate overall model performance of BVOCs over a region. Satellite retrieval of formaldehyde alone is still difficult to constrain uncertain parameters or functions in BVOCs emission scheme, particularly over the regions like China.

Now, we add Fig. 10 and Fig. S7, and the corresponding discussion in the revised manuscript:

[revised manuscript text omitted]

What's more, we are collaborating with some teams who collect BVOCs measurements, such as one of co-authors, Prof. Dasa Gu. As our respond to the reviewer's comments above, modeling results in this study also provide some useful information to the colleagues conducting laboratory experiments and field observations about what, where, and when to measure. We plan to evaluate the model and constrain the emission scheme with more observations of BVOCs emissions and concentrations in future.

- *In terms of specification of vegetation, one can see differences in the vegetation classifications between the dataset and conclude that those will result in different BVOC emissions without even running MEGAN. There should be satellite derived products that could be used to understand differences in the types and spatial distribution of vegetation between observations and datasets used by MEGAN.*

Thanks for your comments. We agree that the difference of vegetation distribution will definitely affect the results to some extent without running the model. However, we always need to conduct numerical experiments to quantify the impacts and understand the difference, which can help us to know where to improve.

In terms of vegetation distribution, in fact, as we discussed in Section 2.3 of manuscript, the VEG-USGS and VEG-2015 datasets are both derived from satellite observations. VEG-USGS is the default land use data in WRF-Chem (v3.6) based on AVHRR satellite product spanning April 1992 through March 1993 using a resolution of 1 km, and VEG-2015 is derived from the MODIS retrievals in 2015, which has the horizontal resolution of 1 km over all of China. Therefore, there is no doubt that VEG-2015 is closer to the reality, particularly for East China with intensive urban expansion since 2000s. One of the purposes of this study is to quantify the impacts of updating the land-use dataset from 1990s

to 2010s. It is necessary because there are still many studies using the default land-use dataset in WRF-Chem for simulations over East China. In addition, it also provides useful information to quantify the impacts of sub-grid vegetation distributions over East China with intensive urban expansion, which is considered in the coupling of MEGANv3.0 but not in that of MEGANv2.0 in WRF-Chem. Lastly but not the least, the analysis of this study also highlights that the "offline" coupling of MEGANv2.0 with prescribed four vegetation types shows negligible sensitivity to the changing vegetation distributions.

Although VEG-2015 should be more representative over East China for 2000s and after than VEG-USGS, it could still have some uncertainties, particularly for the specification of various vegetation types. Field survey and investigation of vegetation types may help us to further improve the land-use dataset used by MEGAN in WRF-Chem for East China.

Now, we add some clarification and discussion in the revised manuscript as:
"Theoretically, VEG-2015 should be more representative for the reality in 2015, particularly for East China with intensive urban expansion since 2000s."
"Although, theoretically, VEG-2015 should be more representative for the reality in 2015, particularly for East China with intensive urban expansion since 2000s, it could still have some uncertainties, particularly for the specification of various vegetation types. The survey of more accurate and higher resolution vegetation distribution based on in-situ survey and investigation should be conducted to support the estimation of BVOC emission over East China."

- *In addition to chemical observations, an evaluation of the predicted surface meteorological quantities (temperature, precipitation, soil moisture, radiation) using observations would have been useful to understand how those uncertainties would influence the predicted BVOCs. A time series of observed and simulated meteorological quantities over the month-long period compared with predicted BVOC emissions would have been useful. The month-long period would also make simulating soil moisture challenging. There are satellite products that could be used to assess soil moisture variability over the region.*

Thanks for your suggestion. We agree that it would be challenging to continuously simulate reasonable meteorology for a month. Therefore, as we mentioned in the manuscript, in this study, the modeled u and v component wind and temperature in atmosphere above the planetary boundary layer are nudged towards the NCEP Final reanalysis data with a 6-hour timescale, which is often used to constrain regional meteorological simulation towards reanalysis. In this way, the key meteorological fields such as winds, temperature, precipitation, radiation are similar among different experiments with different MEGAN versions. The difference in BVOCs emission and concentrations among different experiments can be primarily attributed to the difference in MEGAN versions.

To support this, now we add Fig. S2-S3 in the revised manuscript to show the monthly average of surface meteorological fields (temperature, precipitation, soil moisture and surface net solar radiation) from different simulations with three MEGAN versions in April and July with VEG-2015 and from the FNL reanalysis data. In addition, we add Fig. S4-S5 to illustrate the daily variation of surface meteorological fields over simulation domain in April and July. It is evident that all meteorological fields are consistent in different versions of MEGAN simulations and close to the FNL reanalysis data. Therefore, the simulated meteorological fields are reliable and consistent in order to further investigate the difference and impacts of different MEGAN versions.

Now, we add some clarification and discussion in the revised manuscript as:

"The meteorological initial and lateral boundary conditions are obtained from the NCEP Final reanalysis (FNL) data with 1°×1° resolution and updated every 6 hours. The modeled u and v component wind and temperature in atmosphere above the planetary boundary layer are nudged towards the NCEP Final reanalysis data with a 6-hour timescale (Stauffer and Seaman, 1990). In this way, the simulated key meteorological fields (e.g., surface temperature, precipitation, surface net solar radiation, and soil moisture) are close to the FNL reanalysis data (Fig. S2-S5 in the supporting material), which sets the base for further investigating the impacts of different MEGAN versions. There are a few days when the simulated surface solar radiation fluxes have positive biases, which may be due to the biases of clouds and the ignorance of aerosol radiative impacts in the simulations. The nudged simulations also guarantee that the difference in simulated BVOCs is from the difference in MEGAN versions instead of the meteorological difference."

- *As pointed by the authors, the major difference in MEGAN v3 is the inclusion of a drought activity factor (lines 239-240), but the importance of this factor for the 2 month-long simulations is discussed only briefly (line 486-495) in terms of seasonal changes. The authors do not say whether or not April or July of 2015 are periods characterized by drought. The authors note that the drought factor is constrained by limited observations (line 242) and may not be suitable for China (line 244). So how is one supposed to assess the seasonal impacts of this parameter compared to other parameters in MEGAN (i.e. Figure 7)? Perhaps some sensitivity simulations are needed with a range of values for the drought factor to get a better idea of its potential effect.*

In fact, the primary difference between MEGANv3.0 and previous versions should be the updates to MEGANv2.1 including the different coupling method. In order to keep the version updated, in this study we update MEGANv2.1 to MEGANv3.0 to include the drought activity factor, which does not mean that our focus will be on drought activity factor. We clarify this in Section 2.2 of revised manuscript as:

"The MEGANv3.0 employed in this study is updated from MEGANv2.1 as implemented

by Zhao et al. (2016). Zhao et al. (2016) implemented MEGANv2.1 into CLM4.0 in WRF-Chem so that biogenic emission and land surface processes can use consistent distributions of meteorological fields such as land-use type, surface air temperature, LAI, and solar radiation, which is significantly different from previous versions (v1.0 and v2.0) in terms of scheme structure because the coupling of previous versions of MEGAN is independent of land surface scheme. Compared to the widely used MEGANv2.0 in WRF-Chem that defines emission factor as the total flux of chemical compounds, MEGANv2.1 defines emission factor as the net primary emission that escaped into the atmosphere and it does not contain the downward flux of chemicals from above canopy, as detailed in Zhao et al. (2016). In addition, MEGANv2.1 can also consider sub-grid vegetation distributions, which is different from previous versions that generally apply dominant vegetation type in each grid for BVOC emission calculation. The primary update in MEGANv3.0 from MEGANv2.1 is to consider drought activity factor as one of environmental forcing."
"It is noteworthy that the major difference between MEGANv3.0 and previous versions is primarily caused by the difference between MEGANv2.1 and previous versions as discussed above instead of drought effect."

The drought activity factor is not specific for drought period. It basically reflects the impacts from soil moisture and can take effect in any period. Thanks for your comment about the sensitivity simulations for drought activity factor. Since the coefficient α used for drought activity factor calculation is quite empirical and there is no observations and experiments for constrain drought forcing over East China, we conduct sensitive simulations with a range of values of coefficient α to get a better understanding of its impacts. Now, we add Fig. S6 in the revised manuscript to show the seasonal change of drought activity factor with different α, it is similar as Figure 7 in the manuscript except for different α values. Consistent with the formula (2) in the manuscript, the reduction of α increases the drought activity factor. The seasonal difference increased by ~0.2 over the southern domain. However, overall, the difference between two simulations with different α is not significant. The value of α has small effect on the seasonal variation and the spatial distribution of drought activity over East China. Jiang et al. (2018) also concluded that the drought effect on seasonal change of isoprene emissions in China is not evident.

Now, we discuss about it in the revised manuscript as
"As mentioned in the methodology, the empirical coefficient α of 37 is applied for drought activity factor calculation following Jiang et al. (2018) in this study due to the lack of observation and experiment constraint over China. To examine its potential effect on calculating drought activity factor in China, sensitivity experiments are conducted with different values of α. The results indicate that the value of α has small effect on the seasonal variation and the spatial distribution of drought activity factor over East China (Fig. S6 in the supporting material), which is consistent with Jiang et al. (2018) that also stated the

drought effect on seasonal change of isoprene emissions in China was not evident."

- ***In summary, while I appreciate having differences in the physical representation among the MEGAN versions discussed in one paper, my issues with this paper are 1) the primary main findings have been described previously and 2) there is a lack of any observations needed to fully understand model uncertainty. Section 3.1.3 implies there are observations. If the authors choose to include only modeling analyses, the paper needs to be revised to frame the purpose of the paper better and improve Section 3.1.3 to put their results into the context of what might be observed in the real world, what is needed to assess components of that latest version of MEGAN, and what are the most missing or uncertain treatments – apart from the specification of vegetation.***

Thanks for your constructive suggestions.

(1) As we respond to your comments above, we believe it is necessary to document and demonstrate the difference among different versions because the modeling community never automatically move to the newer version in default. In fact, it is quite common to investigate the difference among different versions of parameterization in the modeling community (i.e., LeGrand et al., 2019; Zhao et al., 2019; Onwukwe and Jackson, 2020; Zeng et al., 2020).

Although the comparison among different versions were explored by Zhao et al. (2016) over California, they focused on emission factors and fluxes only and did not pay much attention on different activity factor functions among different versions. In addition, the WRF-Chem with MEGAN was widely used in China but was not documented well before. Therefore, this study aims to examine the difference among different versions of MEGAN used in WRF-Chem for East China, particularly between MEGANv2.0 and MEGANv3.0 because both of them are still widely used. The comparison of modeling results with different versions of MEGAN can help us understand the mechanisms leading to the difference. Furthermore, we also examine the major changes among different versions of MEGAN under seasonal variation and with different vegetation datasets. This analysis can find the most sensitive parameters (e.g., temperature and leaf area index) that lead to the major difference, which can help us to identify the key processes that needs more accurate observations to constrain. We thus consider our manuscript appropriate for Geoscientific Model Development (GMD).

In addition, although we currently don't have enough observations to evaluate the modeling results, the analysis of difference among different versions can provide useful information about what kind of observations are more urgently needed and over which region of China. For example, this study points out that the sensitivities of biogenic emissions to LAI and temperature are large over China, and thus the laboratory and field experiments are needed to constrain better the functions.

Without enough robust observations to constrain and evaluate the model, we revise our title to "Modeling sensitivities of different versions of MEGAN BVOC emission schemes

in WRF-Chem (v3.6) and its impacts over East China"

(2) Please see our response to your comment above, it will be great if there are some observations for evaluation. However, the direct observations of BVOCs are really scarce over China. Now we clarify the detailed information of collected observations in Table 4 in the revised manuscript. Ideally, we need the observations at multiple sites or from aircraft for a specific period to evaluate overall model performance of BVOCs over a region (e.g., Zhao et al., 2016). Therefore, it is difficult to evaluate effectively any simulations with those observations listed in Table 4.
Now we add the evaluation of simulated total column tropospheric formaldehyde with satellite retrievals in April and July. We add Fig. 10 in the revised manuscript to show the monthly mean total column formaldehyde concentration simulated by different versions of MEGAN in April and July and compared with satellite retrievals. However, due to the comparable contributions to formaldehyde from both anthropogenic and biogenic sources over East China and the uncertainties of satellite retrievals and anthropogenic emissions, satellite retrieval of formaldehyde alone is still difficult to constrain uncertain parameters or functions in BVOCs emission scheme, particularly over the regions like China. More direct observations of BVOCs at multiple sites or from aircraft for different seasons are needed to evaluate overall model performance of BVOCs over East China.

Now we add the clarification in the revised manuscript as "In summary, this study documents the different performance among different versions of MEGAN and its impacts on ozone and other chemical compounds, which can provide the WRF-Chem community more comprehensive analysis to understand the mechanisms of modeling sensitivities in BVOCs among different versions of MEGAN in WRF-Chem and vegetation distributions. The different response to seasonal change and vegetation distributions are also quantified. On the other hand, modeling sensitivity analysis also provides more information about what and where to measure for better constraining the modeling of BVOCs over East China."

[revised manuscript text omitted]

- ***In addition, the abstract has numerous awkward phrases and I pointed out those in the specific comments below. However, I did not point out other instances throughout the paper.***

We appreciate that you point out awkward phrases. We checked and corrected these errors in the revised manuscript.

*Specific Comments:*

- ***Line 37: The phrase "over East China" seems out of place and could be deleted. The first part of the sentence actually applies to many places in the world, not just East China. The authors should true to put the paper into a context for a broad range of readers whenever possible.***

Thanks for your suggestions. Now the sentences are revised as "Biogenic volatile organic compounds (BVOCs) simulated by current air quality and climate models still have large uncertainties, which can influence atmosphere chemistry and secondary pollutant formation."

- ***Lines 37-38: Change "are generally resulted from", which is an awkward phrase, to "are primarily due to".***

Corrected as suggestion.

- ***Lines 38-39: Change "in model" to "in the model".***

Corrected as "in the model".

- ***Line 38-40: Change this sentence to "One originates from different treatments in the physical and chemical processes associated with the emission rates of BVOCs."***

Thanks for your suggestion. Corrected.

- *Line 40: Change "from the biased distribution of vegetation types" to "errors in the specification of vegetation types and their distribution".*

Corrected as suggestion.

- *Lines 47-48: Change "biogenic VOCs" to BVOCs which was already defined.*

Corrected.

- *Line 59: The authors use "significantly" here, but that is a vague term. Can they provide a number to quantify range?*

Now we provide a range to quantify the difference in the revised manuscript as "The simulated surface ozone concentration due to BVOCs can be significantly different (ranging from 1 ppbv to more than 10 ppbv in some regions) among the experiments with three versions of MEGAN, which is mainly due to their impacts on surface VOCs and NOx concentrations".

- *Lines 61-62: This sentence just states that the three versions of MEGAN produced different results, which is not surprising since updates usually include new treatments based on observations or theoretical relationships. Presumably, older versions have out-of-date information and should be less accurate. But at this point, the abstract fails to say anything about how the simulated BVOCs compares to observations which is a better way to characterize uncertainty.*

As our response to the reviewer's comment above, this study defines "uncertainty" as the modeling difference among different versions of MEGAN parameterization. This may be mis-leading in different communities. Therefore, we change "uncertainty" to "sensitivity" through the revised manuscript. The title of manuscript is also revised as "Modeling sensitivity of BVOC to different versions of MEGAN schemes in WRF-Chem(v3.6) and its impacts over East China".

This sentence is removed, and we add more discussion about the comparison with observations here in the revised manuscript as:
"However, considering uncertainties of retrievals and anthropogenic emissions over East China, it is still difficult to apply satellite retrievals of formaldehyde and/or limited sparse in-situ observations to constrain the uncertain parameters or functions in BVOCs emission schemes and their impacts on photochemistry and ozone production. More accurate vegetation distribution and measurements of biogenic emission fluxes and species concentrations are still needed to better evaluate and optimize models."

- *Lines 166-182: This section does not list the land-surface parameterization used. As noted by other studies MEGAN2.1 (called here MEGAN3.0) is coupled to the CLM*

*land surface parameterization. The other versions of MEGAN can use different land-surface parameterizations. So it is not clear whether differences in the land-surface treatment, which will affect the surface energy budget and fluxes, will affect the BVOC emissions. While Zhao et al. (2016) found that differences due to the land-surface parameterization were of secondary importance compared to specification of the vegetation, that point on the model configuration should be cleared up. Also, I assume that meteorological predictions should be identical among the simulations? That should be explicitly stated.*

Thanks for your suggestion. All simulations in this study used CLM4 land surface parameterization. Now we list the WRF-Chem configurations for all simulations in Table 1 in the revised manuscript.

Please see our response to the reviewer's comment above, the meteorological initial and lateral boundary conditions of all simulations are obtained from the NCEP Final reanalysis data with $1°×1°$ resolution, and the modeled u and v component wind and temperature in the atmosphere above the planetary boundary layer are nudged towards the NCEP Final reanalysis data with a 6-hour timescale. In this way, the key meteorological fields such as winds, temperature, precipitation, radiation are similar among different experiments with different MEGAN versions. The difference in BVOCs emission and concentrations among different experiments can be primarily attributed to the difference in MEGAN versions. Now we add some comparison and evaluation of meteorological fields in different experiments in the supporting material.

- *Lines 512-513: This is a very general statement. Which results are the authors talking about? BVOC concentrations or some other quantities?*

Thanks for your suggestion. The results are the simulated isoprene concentrations. The sentence is revised as "In general, the simulated isoprene concentrations from MEGANv2.0 and MEGANv3.0 are closer to measurements in these four sites listed in Table 4.".

- *Lines 514-515: This statement needs to be backed up by the evidence that is not shown.*

It is difficult to obtain the uncertainty information about the observations based on the literatures for these limited and scattered observations. Now we delete this statement and add more discussion in the revised manuscript as "Compared with the limited observations, MEGANv2.0 produces higher isoprene concentrations in most sampling sites except the site of Lishui-District surrounded with the densely vegetation-covered suburb. As discussed above, MEGANv3.0 can simulate higher biogenic isoprene emissions in urban area due to its consideration of sub-grid vegetation distributions, At the sampling sites in urban area, such as the sites of Xujiahui, Shanghai, and Nanjing, the simulation with MEGANv3.0 produces higher surface concentration of isoprene compared to that with

MEGANv1.0. In MEGANv2.0, the prescribed vegetation distributions do not reflect the urbanization over East China. Therefore, the simulated isoprene concentrations between these two versions are comparable. Overall, the experiments with MEGANv2.0 and MEGANv3.0 may simulate better surface concentration of isoprene over East China than that with MEGANv1.0. Please note that these observations were collected at different sites for different periods. Ideally, the observations at multiple sites or from aircraft for a specific period are needed to evaluate overall model performance of BVOCs over a region (e.g., Zhao et al., 2016). It is difficult to evaluate effectively any simulations with those observations listed in Table 4."

In addition, we add more discussion about the comparison with satellite retrievals in the revised manuscript. Please see our response to the reviewer's comment above.

- ***Lines 583: There can be anthropogenic isoprene emissions so how did the author separate out the biogenic source of VOC concentrations versus the anthropogenic sources?***

Sorry for the confusion. The concentrations of species contributed by biogenic emissions are estimated through calculating the difference between the control simulation and the simulation without biogenic emissions. Now it is clarified in Section 3.2 of the revised manuscript as "The concentrations of species contributed by biogenic emissions are estimated through calculating the difference between the control simulation and the simulation without biogenic emissions.".

- ***Line 551: The authors need to state how the calculation of ozone from biogenic and anthropogenic sources are separated out. I assume this is a highly non-linear system and it is not easy to separate out these sources since biogenic emissions will influence the anthropogenic system and visa versa as emission sources mix. This comment also applies to NOx sources in lines 562-563.***

Thanks for your comment. See our response to the comment above. The concentrations of species contributed by biogenic emissions are estimated through calculating the difference between the control simulation and the simulation without biogenic emissions. Therefore, in this way, the non-linear feedback of biogenic emissions is also considered. The impacts of biogenic emission include both its direct contribution to VOCs concentrations but also its impacts on the photochemical system.

- ***Line 580: At the end of Section 3, there needs to be some discussion regarding the meteorological conditions during the simulation periods in 2015. Some the figures (6,7,9) plot differences between July and April. Presumably the seasonal changes are dependent on meteorology which might change from year to year. The authors should discuss whether these differences would be representative of other years and why.***

Thanks for your comment to raise an important point. Following your suggestion, we

conducted another set of experiments for 2016, and analyze the results for Fig. 6, 7, 9 for 2016. As shown in Fig. R1-3, the results between 2015 and 2016 are quite similar. Therefore, we can generally conclude that the results and conclusion in this study does not change significantly for different year.

Now we add some discussion in the revised manuscript as "Although the analysis in this study is for one single year, the investigation of simulations for a different year demonstrates similar results (not shown), which indicate the modeling sensitivities with different versions of MEGAN do not change significantly with years."

[Figure]

**Figure R1.** Spatial distribution of the quotient of biogenic isoprene emission and activity factor between simulations in July and that in April in 2016, using VEG-2015 vegetation data set.

[Figure]

**Figure R2**. Spatial distribution of the quotient of activity factor related to different environmental variables (such as temperature, LAI, light, leaf age and drought) between

simulations in July and that in April (July divided by April) in 2016 using VEG-2015 vegetation data set coupled with MEGAN v3.0.

[Figure]

**Figure R3.** The spatial distribution of monthly leaf area index (LAI) in 2016 simulated by MEGAN v3.0.


and Fu, T. M.: Influences of planetary boundary layer mixing parameterization on summertime surface ozone concentration and dry deposition over North China, Atmospheric Environment, 218, 9, 2019.

**Anonymous Referee #3**

*General comments:*

- *Adding the capability of MEGAN v3.0 into WRF-chem is a great advancement for the community. However, the paper needs to be restructured and more evaluation of the model updates against observations are needed. The main problem is that the overall conclusions of the paper (there are large uncertainties in the BVOC emission schemes) are not supported by the results of the paper.*

We thank the reviewer for the detailed and constructive comments. They are very helpful for improving the quality of the manuscript. First of all, to respond to your comment, this study defines "uncertainty" as the modeling difference among different versions of MEGAN parameterization. This may be mis-leading in different communities. Therefore, we change "uncertainty" to "sensitivity" through the revised manuscript. The title of manuscript is also revised as "Modeling sensitivities of BVOCs to different versions of MEGAN emission schemes in WRF-Chem (v3.6) and its impacts over East China".

- *Differences between sequential updates of MEGAN v1, v2, and v3 does not demonstrate uncertainty as presumably the later versions are more accurate as explained further below. The paper could easily be restructured to show how the sequential updates of MEGAN v1, v2, and v3, which increase in process complexity, have led to improvements (or not) in BVOC emissions, ozone, and PM2.5. This restructuring should better emphasize the isoprene observations specified in Table 3 and compare to other observations such as ozone and PM2.5 that may be more readily available than isoprene as explained more below.*

Sorry for the confusion. Now we revise the manuscript to focus on modeling sensitivities instead of evaluate modeling uncertainties.

It is often expected that the updates in the newer version can improve the model performance, however, it is not always true as we all know. Therefore, it is necessary to document and demonstrate the difference among different versions because the modeling community never automatically move to the newer version in default. In fact, it is quite common to investigate the difference among different versions of parameterization in the modeling community (i.e., LeGrand et al., 2019; Zhao et al., 2019; Onwukwe and Jackson, 2020; Zeng et al., 2020).

The WRF-Chem with MEGAN was widely used in China but was not documented well before. Therefore, this study aims to examine the difference among different versions of MEGAN used in WRF-Chem for East China, particularly between MEGANv2.0 and MEGANv3.0 because both of them are still widely used. The comparison of modeling results with different versions of MEGAN can help us understand the mechanisms leading to the difference. Furthermore, we also examine the major changes among different versions of MEGAN under seasonal variation and with different vegetation datasets. This

analysis can find the most sensitive parameters (e.g., temperature and leaf area index) that lead to the major difference, which can help us to identify the key processes that needs more accurate observations to constrain. We thus consider our manuscript appropriate for Geoscientific Model Development (GMD).

In the revised manuscript, we also add more comparison and discussion with different types of observations. Although we do have observations of ozone and NOx at multiple sites over East China, it is difficult to evaluate the simulated BVOCs with those observations. First, NOx and ozone can be largely affected by NOx emission that is quite uncertain over East China. Therefore, it is hard to indirectly evaluate the simulated BVOCs with NOx and ozone. Second, although ozone can be largely affected by VOCs and can reflect the strength of VOCs to some extent, VOCs are contributed comparably by anthropogenic and biogenic VOC over East China, which is not like some regions such as the southeastern United States and the Amazon area where VOCs are dominated by BVOCs. Formaldehyde is often used to reflect the overall burden of VOCs. Now we add Fig. S7 in the revised manuscript to show the spatial distribution of anthropogenic and biogenic column formaldehyde concentration near the surface in April and July using VEG-2015. It is evident that anthropogenic and biogenic formaldehyde concentrations are comparable over most regions of East China. On the whole, it is difficult to use observations of ozone and NOx to evaluate the simulated BVOCs over East China.

In fact, we also thought about using satellite retrieval of formaldehyde to indirectly evaluate the simulated BVOCs. However, as we discussed above, we concerned about the difficulties to isolate the contributions from anthropogenic and biogenic BVOCs in retrievals, therefore we didn't show the comparison in the original manuscript. Upon the reviewers' suggestion, now we evaluate the simulated total column tropospheric formaldehyde with satellite retrievals in April and July. We add Fig. 10 in the revised manuscript to show the monthly mean total column formaldehyde concentration simulated by different versions of MEGAN in April and July and compared with satellite retrievals. In general, the simulated tropospheric column formaldehyde concentrations are consistent with satellite retrievals in April, showing high column formaldehyde concentration over the Yangtze River Delta region and South China. Based on Fig. 10 and Fig. S7, the formaldehyde concentrations are contributed comparably by both anthropogenic and biogenic sources over these two regions in July, and biogenic source contributes about 20% to the total in April. Although there are some small differences in formaldehyde column concentrations in April among the simulations with different MEGAN versions, consistent with the comparison of biogenic emissions (Fig. 3), it is difficult to apply the satellite retrievals to constrain their small difference if considering the uncertainties of retrievals. In July, the difference among the simulations with different MEGAN versions is much larger. Compared to the satellite retrievals, the simulation with MEGANv1.0

(MEGANv3.0) may underestimate (overestimate) tropospheric formaldehyde column concentrations. These biases may reflect their errors in biogenic emissions. The large difference between MEGANv2.0 and MEGANv3.0 in July may indicate that some activity factors controlling seasonal variation of BVOCs emissions is less appropriate in MEGANv3.0 than in MEGANv2.0. However, please note that satellite retrievals of formaldehyde may also have relatively large uncertainties in July (e.g., Su et al., 2019; Su et al., 2020) and the uncertainties of anthropogenic emissions of VOCs may also contribute to the modeling biases of formaldehyde. As we mentioned above, more direct observations of BVOCs at multiple sites or from aircraft for different seasons are needed to evaluate overall model performance of BVOCs over a region. Satellite retrieval of formaldehyde alone is still difficult to constrain uncertain parameters or functions in BVOCs emission scheme, particularly over the regions like China.

Now we add Fig. 10 and Fig. S7, and the corresponding discussion in the revised manuscript:

[revised manuscript text omitted]

- *Also as further explained below related to Figure 12, please confirm that soil NOx does not change between your MEGAN v1, v2, and v3 configurations?*

Soil NOx is not included in this study. Now we clarify this in the revised manuscript as "Soil and lightning NOx sources are not included in this study."

*Specific comments:*
- *Line 35 and 61: Because the differences in BVOC emissions from MEGAN v1, v2, and v3 are sequentially improving, the differences between the versions do not demonstrate uncertainty in BVOC emissions, but instead model advances. Certainly, there is still a lot of uncertainty in BVOC emissions, but this paper does not really address these uncertainties. For example, testing the uncertainty in the MEGAN inputs like emission factors and vegetation types or comparing MEGAN to an entirely different BVOC emission scheme would be a way to evaluate uncertainty, but comparing MEGAN to older versions of itself without a more thorough comparison against observations than that provided in this work is not an appropriate way to demonstrate uncertainty.*

Sorry for the confusion. This study defines "uncertainty" as the modeling difference among different versions of MEGAN parameterization. This may be mis-leading in different communities. Therefore, we change "uncertainty" to "sensitivity" through the revised manuscript and revise these two sentences. The title of manuscript is also revised as "Modeling sensitivities of BVOCs to different versions of MEGAN emission schemes in WRF-Chem (v3.6) and its impacts over East China". Please also see our response to the comments above.

- *Line 153, Please add a reference for MEGAN v3.0 here.*

The reference is added now.

- *Line 211: Please be very clear in this section how your version of MEGAN v2.0 differs from the released version. For example, state what is available in the released version and then specify which options you use or adjust in this work.*

Thanks for your suggestion. The MEGANv2.0 in WRF-Chem used in this study is the default one in the publicly released version 3.6 of WRF-Chem, and we did not modify MEGANv2.0 in WRF-Chem. It is different from the offline MEGANv2.0. For example, The emission factors of BVOCs for each grid cell can be prescribed or calculated with prescribed vegetation distribution and emission factor for each PFT in offline MEGANv2.0, while the emission factor of isoprene is prescribed at each grid cell from an input dataset in the online MEGANv2.0 in WRF-Chem. To clarify this, we add the discussion in the revised manuscript as "Guenther et al. (2006) introduced MEGANv2.0 that is a major update from the previous version. The emission factor of BVOCs for each grid cell can be prescribed or calculated with prescribed vegetation distribution and emission factor for each PFT. For activity factor, the impacts of PPFD, temperature, monthly LAI, leaf age, soil moisture, and solar radiation on biogenic emissions are taken into account (Guenther

et al., 2006). Different from the released offline MEGANv2.0, after coupled with WRF-Chem, MEGANv2.0 reads emission factor at each grid cell for isoprene and calculate emission factors for other BVOCs based on PFT's and PFT-specified emission factor at each grid cell. The vegetation distribution at each grid cell used in MEGANv2.0 in WRF-Chem includes only 4 dominant PFT at each grid cell and is prescribed differently from the one used in the land surface scheme (e.g., 24 land-use types). In addition, the MEGANv2.0 coupled with the publicly released versions of WRF-Chem uses the monthly mean surface air temperature, LAI and solar radiation from the climatological database that may not be consistent with the meteorological fields during simulation.".

- *Line 243: Is there an estimate on the uncertainty on this alpha value? Some sensitivity tests showing the range of possibilities of this alpha value and its impact on isoprene would demonstrate existing uncertainty in BVOC emission schemes.*

Thanks for your comment about the sensitivity simulations for drought activity factor. Since the coefficient α used for drought activity factor calculation is quite empirical and there is no observations and experiments for constrain drought forcing over East China, we conduct sensitive simulations with a range of values of coefficient α to get a better understanding of its impacts. Now, we add Fig. S6 in the revised manuscript to show the seasonal change of drought activity factor with different α. It is similar as Figure 7 in the manuscript except for different α values. Consistent with the formula (2) in the manuscript, the reduction of α increases the drought activity factor. The seasonal difference increased by ~0.2 over the southern domain. However, overall, the difference between two simulations with different α is not significant. The value of α has small effect on the seasonal variation and the spatial distribution of drought activity over East China. Jiang et al. (2018) also concluded that the drought effect on seasonal change of isoprene emissions in China is not evident.

Now, we discuss about it in the revised manuscript as
"As mentioned in the methodology, the empirical coefficient α of 37 is applied for drought activity factor calculation following Jiang et al. (2018) in this study due to the lack of observation and experiment constraint over China. To examine its potential effect on calculating drought activity factor in China, sensitivity experiments are conducted with different values of α. The results indicate that the value of α has small effect on the seasonal variation and the spatial distribution of drought activity factor over East China (Fig. S6 in the supporting material), which is consistent with Jiang et al. (2018) that also stated the drought effect on seasonal change of isoprene emissions in China was not evident."

- *Line 262: It appears this VEG-2015 is a land cover map calculated in this work. Has this or a similar approach been used in other papers even if for different regions of the world or for different models? If so, please reference them. If not, please include*

*more references on the advantages to using this new approach to make it clearer to the reader, which land cover type is better.*

VEG-USGS and VEG-2015 are both satellite products. VEG-USGS is the built-in land use data in WRF-Chem based on AVHRR satellite data spanning April 1992 through March 1993 at the resolution of 1 km. VEG-2015 is derived from the MODIS retrievals in 2015, which has the horizontal resolution of 1 km over China. VEG-2015 is derived specifically for this study and has not been published before, and it is converted to 16-PFT data set in CLM4.0 following the same method as the VEG-USGS. Theoretically, VEG-2015 is better than VEG-USGS for the simulations of 2015 in this study, because it can better reflect the rapid urbanization over East China after 2000s. Now we add the clarification in the revised manuscript as: "Theoretically, VEG-2015 should be more representative for the reality in 2015, particularly for East China with intensive urban expansion since 2000s."

- *Figure 4: Why is v2.0 here not separated by land cover type like v1.0 and v3.0 in this figure? And which land cover type is used here for v2.0?*

As we clarify in the revised manuscript now (see our response to the reviewer's comment above), different from the released offline MEGANv2.0, after coupled with WRF-Chem, MEGANv2.0 reads emission factor at each grid cell for isoprene. Therefore, the emission factor of isoprene for MEGANv2.0 in WRF-Chem is independent on vegetation distributions. For other species of BVOCs, the emission factor is calculated with the PFT-specified emission factor and prescribed spatial distributions of four dominant PFT's, which is also independent on vegetation distributions. Now we add the clarification in the caption of Fig. 4 as "Please note the emission factors of isoprene are prescribed for MEGANv2.0 in WRF-Chem and therefore are independent on vegetation distributions."

- *Figure 5: For clarity, please list what PFT-1, -4, and -6 refer to in the figure or figure caption.*

Thanks for your suggestion. Now, the descriptions about PFT's are added in the caption of Fig. 5.

- *Line 414: Which MEGAN versions do you mean by "current versions of MEGAN"*

Clarified as "in the three versions of MEGAN in WRF-Chem."

- *Line 449: This section is quite interesting and demonstrates important improvements to MEGAN v3.0 versus v2.0 and why future model simulations should move to MEGANv3.0. Seems like these advances and results should be highlighted more clearly in the overall conclusions.*

Thanks for your suggestion. Now we highlight this in the abstract and conclusion as "Theoretically MEGANv3.0 that is coupled with the land surface scheme and considers the sub-grid vegetation effect should overcome previous versions of MEGAN in WRF-Chem, However, considering uncertainties of retrievals and anthropogenic emissions over

East China, it is still difficult to apply satellite retrievals of formaldehyde and/or limited sparse in-situ observations to constrain the uncertain parameters or functions in BVOCs emission schemes and their impacts on photochemistry and ozone production. More accurate vegetation distribution and measurements of biogenic emission fluxes and species concentrations are still needed to better evaluate and optimize models."
, and
"Theoretically MEGANv3.0 that is coupled with the land surface scheme and considers the sub-grid vegetation effect should overcome previous versions of MEGAN in WRF-Chem, however, considering uncertainties of retrievals and anthropogenic emissions over East China, limited in-situ observations or satellite retrieval of formaldehyde alone is still difficult to constrain uncertain parameters or functions in BVOCs emission schemes applied over East China. High-quality direct observations of BVOCs emissions or concentrations for different season at multiple sites or from aircrafts in both rural and urban areas of East China are definitely needed to evaluate overall model performance of BVOCs over China, particularly over some specific areas with large modeling sensitivities of BVOC emission and activity factors, such as the Anhui and Henan provinces in the north of simulation domain, suggested by this study. In addition, direct measurements of biogenic emission fluxes and/or emission factors and activity factors in the laboratory may be also helpful to constrain different activity factor functions of MEGAN in atmospheric models."

- *Line 507: Please provide references here. From Figure 10, the formaldehyde concentration looks to be quite different between the MEGAN versions especially in the northern part of the domain. These are not large enough differences to see on a satellite? Plotting formaldehyde from anthropogenic VOC emissions here would be useful to fully explain this.*

Please also see our response to the reviewer's comment above.

Now we evaluate the simulated total column tropospheric formaldehyde with satellite retrievals in April and July. We add Fig. 10 in the revised manuscript to show the monthly mean total column formaldehyde concentration simulated by different versions of MEGAN in April and July and compared with satellite retrievals. In general, the simulated tropospheric column formaldehyde concentrations are consistent with satellite retrievals in April, showing high column formaldehyde concentration over the Yangtze River Delta region and South China. Based on Fig. 10 and Fig. S7, the formaldehyde concentrations are contributed comparably by both anthropogenic and biogenic sources over these two regions in July, and biogenic source contributes about 20% to the total in April. Although there are some small differences in formaldehyde column concentrations in April among the simulations with different MEGAN versions, consistent with the comparison of biogenic emissions (Fig. 3), it is difficult to apply the satellite retrievals to constrain their small difference if considering the uncertainties of retrievals. In July, the difference among the simulations with different MEGAN versions is much larger. Compared to the satellite retrievals, the simulation with MEGANv1.0 (MEGANv3.0) may underestimate (overestimate) tropospheric formaldehyde column concentrations. These biases may reflect their errors in biogenic emissions. The large difference between MEGANv2.0 and

MEGANv3.0 in July may indicate that some activity factors controlling seasonal variation of BVOCs emissions is less appropriate in MEGANv3.0 than in MEGANv2.0. However, please note that satellite retrievals of formaldehyde may also have relatively large uncertainties in July (e.g., Su et al., 2019; Su et al., 2020) and the uncertainties of anthropogenic emissions of VOCs may also contribute to the modeling biases of formaldehyde. As we mentioned above, more direct observations of BVOCs at multiple sites or from aircraft for different seasons are needed to evaluate overall model performance of BVOCs over a region. Satellite retrieval of formaldehyde alone is still difficult to constrain uncertain parameters or functions in BVOCs emission scheme, particularly over the regions like China.

- *Table 3: Please provide significantly more detail about the observations listed in Table 3. For example, are these observations averaged over some time period? Is the averaging in the model and observations the same? When (year, season, month) were the observations collected? If the time was different than the model runs, how would this impact the comparison? For the first observation, this is no reference listed. Please add more info about where this data came from.*

Thanks for the comments. As you are also aware, it is difficult to compare the simulation with the limited sparse observations. These observations are at different sites and for different time period. All of these observation values are the mean over the entire time period. There is no way to match simulations and observations for comparison. Now we add more detailed information to each observation in Table 4 of the revised manuscript.

Now we add more discussion about the comparison with these limited observations. Please see our response to the reviewer's comment above.

- *Line 533: Please explain in more detail how you got these biogenic VOC and biogenic formaldehyde values. Did you run the model without anthropogenic VOC emissions or without anthropogenic VOC and NOx emissions or did you use a tag for formaldehyde production from biogenic VOCs? This is important to describe as anthropogenic NOx will impact the production rate of formaldehyde from BVOC emissions.*

Sorry for the confusion. The concentrations of species contributed by biogenic emissions are estimated through calculating the difference between the control simulation and the simulation without biogenic emissions. Therefore, in this way, the non-linear feedback of biogenic emissions is also considered. The impacts of biogenic emission include both its direct contribution to VOCs concentrations but also its impacts on the photochemical system. Now it is clarified in Section 3.2 of the revised manuscript as "The concentrations of species contributed by biogenic emissions are estimated through calculating the difference between the control simulation and the simulation without biogenic emissions.".

- *Line 535: Adding similar plots of the total VOC and total formaldehyde would make*

*it clearer that BVOCs contribute significantly to total VOCs over East China. This statement is contradictory to your statement previously for why you could not use satellite formaldehyde to evaluate the changes in BVOC emissions, please explain further.*

Thanks for your suggestion. Now we add Fig. S7 to show the contributions to total column formaldehyde concentrations from anthropogenic and biogenic sources. Compared to Fig. 10, the figure shows that BVOCs contribute significantly (35% on average) to the amount of formaldehyde over East China, which is comparable to the contribution from anthropogenic sources. We believe satellite retrievals of formaldehyde may only be used for effective evaluation if anthropogenic or biogenic emission is the dominant source (>90%). Now we clarify this in the revised manuscript as "It is evident that BVOCs contribute significantly, 25% and 35% on average, to the amount of total VOCs and formaldehyde, respectively, over East China, and the difference among the simulations with the three versions of MEGAN is large (Fig. S7).".

- *Figure 11: These are large differences in surface ozone. Please describe more what is meant by "ozone due to biogenic emissions" is this the ozone value for the simulation with all emissions on minus that without anthropogenic emissions? And in the simulation without anthropogenic emissions does this include removing NOx and VOC emissions? Because the combination of anthropogenic NOx and biogenic VOCs leads to ozone production. It is important to be very clear here what you mean by ozone due to biogenic emissions. Is this biogenic VOCs only or is this biogenic VOCs and anthropogenic NOx? Most of the model domain is maxed out in MEGAN v3.0 please increase the range of this color bar. Also given the large impact these MEGAN inventories have on surface ozone and possibly surface PM2.5 through SOA formation, comparisons against surface ozone and PM2.5 observations would be useful for evaluating the updates to the model. Since there are not a lot of isoprene measurements available, evaluating ozone and PM2.5 is a good second choice.*

As our response to your comment above, the concentrations of species contributed by biogenic emissions are estimated through calculating the difference between the control simulation and the simulation without biogenic emissions. Therefore, in this way, the non-linear feedback of biogenic emissions is also considered. The impacts of biogenic emission include both its direct contribution to VOCs concentrations but also its impacts on the photochemical system. Now it is clarified in Section 3.2 of the revised manuscript as "The concentrations of species contributed by biogenic emissions are estimated through calculating the difference between the control simulation and the simulation without biogenic emissions.".

The color-bar of Fig. 11 is adjusted to show the large values from MEGANv3.0.

Although BVOCs can have large impacts on NOx, ozone, and PM2.5, it is still difficult to

evaluate the simulated BVOCs with those observations. First, NOx and ozone can be largely affected by NOx emission that is quite uncertain over East China. Therefore, it is hard to indirectly evaluate the simulated BVOCs with NOx and ozone. Second, although ozone can be largely affected by VOCs and can reflect the strength of VOCs to some extent, VOCs are contributed comparably by anthropogenic and biogenic VOC over East China, which is not like some regions such as the southeastern United States and the Amazon area where VOCs are dominated by BVOCs. It is the same issue for using the observations of PM2.5. Considering large uncertainties in production of secondary organic aerosol from VOCs, it is even more difficult to use PM2.5 to evaluate the BVOC emission scheme. Overall, high-quality direct observations of BVOCs emissions or concentrations for different season at multiple sites or from aircrafts in both rural and urban areas of East China are needed to evaluate overall model performance of BVOCs over the region and constrain the modeling sensitivities.

- *Line 568 and Figure 12: Does soil NOx change at all in these simulations between MEGAN v1.0, v2.0, and v3.0 or do you turn soil NOx off in these simulations? The differences in Figure 12 look more like differences in soil NOx than changes in the NOx lifetime? If soil NOx is different between these simulations, please calculate the total soil NOx emitted and make sure the values seem reasonable compared to other studies.*

Soil NOx is not included in this study. Now we clarify this in the revised manuscript as "Soil and lightning NOx sources are not included in this study."

- *Line 644: I do not agree that differences in BVOC emissions calculated by MEGAN v3.0 compared to older versions of MEGAN especially very old versions like v1.0 that are not used in any current model demonstrates uncertainty. See first comment. Restructuring this paper toward evaluating the differences in the MEGAN versions and determining whether MEGAN v3.0 improves (or not) model performance of BVOCs, ozone, and PM2.5 would be more useful.*

Thanks for your comment. This sentence is removed.

As our response to the reviewer's comment above, we change "uncertainty" to "sensitivity" through the revised manuscript. The title of manuscript is also revised as "Modeling sensitivities of BVOCs to different versions of MEGAN emission schemes in WRF-Chem (v3.6) and its impacts over East China".

We add a lot more discussion about the evaluation with available observations and what we can learn from the modeling sensitivity analysis. Overall, high-quality direct observations of BVOCs emissions or concentrations for different season at multiple sites or from aircrafts in both rural and urban areas of East China are needed to evaluate overall model performance of BVOCs over the region and constrain the modeling sensitivities.

*Technical corrections:*

- ***-In key point 4, do you mean "is sensitive to the MEGAN version"?***

Yes. Now the sentence is corrected as "but the impact is sensitive to the MEGAN versions."

- ***-Line 39 "coupled in the model"***

Revised as "One originates from different treatments in the physical and chemical processes associated with the emission rates of BVOCs. The other is errors in the specification of vegetation types and their distribution over a specific region."

- ***-Line 39, do you mean "chemical processes" and I would restructure this sentence to be clearer.***

We revise the whole sentence as our response to the comment above.

- ***-Line 74 "VOCs play"***

Corrected as suggestion.

- ***-Line 97 "the impact of BVOCs on air pollutants" or rephrase in another way***

Corrected as suggestion.

- ***-Line 150 "to investigate"***

Corrected as suggestion.

- ***-Line 194 "loss and production"***

Corrected as suggestion.

- ***-Line 263 "over all of China"***

Corrected as suggestion.

- ***-Line 309 Remove extra spaces and "emissions are"***

Corrected as suggestion.

- ***-Line 311: "More details"***

Corrected as suggestion.

- ***-Line 336: "USGS, large differences"***

Corrected as suggestion.

- ***-Line 379 "Over the southwest domain"***

Corrected as suggestion.

- ***-Line 395 "when estimating"***

Corrected as suggestion.

- ***-Line 475 and 478 "Light-dependent"***

Corrected as suggestion.

- ***-Line 504: "for use in this study"***

Corrected as suggestion.

- ***-Line 588: "MEGAN v3.0 includes"***

Corrected as suggestion.

- ***-Line 595: "Physical and chemical"***

Corrected as suggestion.

- ***-Line 1201 "dataset"***

Corrected as suggestion.

---

## Author Response (AR2)

*General comments:*

- *The authors have addressed my comments well. I agree with how the text has been updated to use sensitivities instead of uncertainty when comparing different MEGAN versions. Also I like how you have updated the descriptions of the MEGAN versions as this is much clearer. I also like how you have added comparisons against formaldehyde satellite columns and incorporated more information for all the ground isoprene observations in Table 4. Thanks for making these update.*

We thank the reviewer for the detailed and valuable comments through the review process. These suggestions have guided us to make corresponding corrections and supplements very well, and they are very helpful for improving the quality and fluency of the manuscript.

*Technical corrections:*

- *Line 116: It is okay to use "large uncertainties" here instead of "large modeling sensitivities" if you desire. Here you are directly explaining the current uncertainties in estimating BVOC emissions using MEGAN, so it is fair to use uncertainty here.*

Yes. We revised the sentence as "However, there still remain larger uncertainties in the estimation of BVOCs emission with MEGAN, due to the uncertain emission rates of some compounds, the limited knowledge of environmental activity factors controlling the BVOCs emissions, the accuracy of vegetation distributions, and etc. (Guenther, 2013)."

- *Line 462: "southwest of domain" to "southwest region of the domain"*

Corrected.

- *Line 509: "north of domain" to "northern part of the domain"*

Corrected.

- *Line 801-803: This sentence is kind of a clunky: Possibly rephrase to something like this: "BVOC emissions over East China are sensitive to the version of MEGAN used."*

We rephrased the sentence as "The BVOCs emissions over East China are sensitive to the versions of MEGAN used".

- *Line 803 for "previous studies" here, can you provide a reference?*

We cited references here.

- *Line 1713: "dataset" and do you mean "the first column is from the satellite retrievals"?*

We corrected the figure caption as "Spatial distributions of total column tropospheric formaldehyde concentration (include biogenic and anthropogenic emissions) in April

and July with different versions of MEGAN using the VEG-2015 vegetation dataset. The first column is from the satellite retrievals.".